# Principles of cellular resource allocation revealed by condition-dependent proteome profiling

**Eyal Metzl-Raz[†], Moshe Kafri[†], Gilad Yaakov, Ilya Soifer, Yonat Gurvich, Naama Barkai***

Department of Molecular Genetics, Weizmann Institute of Science, Rehovot, Israel

**Abstract** Growing cells coordinate protein translation with metabolic rates. Central to this coordination is ribosome production. Ribosomes drive cell growth, but translation of ribosomal proteins competes with production of non-ribosomal proteins. Theory shows that cell growth is maximized when all expressed ribosomes are constantly translating. To examine whether budding yeast function at this limit of full ribosomal usage, we profiled the proteomes of cells growing in different environments. We find that cells produce excess ribosomal proteins, amounting to a constant $\approx 8\%$ of the proteome. Accordingly, $\approx 25\%$ of ribosomal proteins expressed in rapidly growing cells does not contribute to translation. Further, this fraction increases as growth rate decreases and these excess ribosomal proteins are employed when translation demands unexpectedly increase. We suggest that steadily growing cells prepare for conditions that demand increased translation by producing excess ribosomes, at the expense of lower steady-state growth rate.

DOI: https://doi.org/10.7554/eLife.28034.001

**\*For correspondence:**
naama.barkai@weizmann.ac.il

[†]These authors contributed equally to this work

## Introduction

Producing ribosomes is a major biosynthetic process in cells. Rapidly growing budding yeast, for example, contain about 190,000 ribosomes (*von der Haar, 2008*), each of which requires the production of over a hundred different proteins. Overall, $\approx 30\%$ of the proteome of rapidly growing cells encodes for ribosomal proteins and $\approx 80\%$ of all cellular RNA encodes for the rRNA components of the ribosomes (*Warner, 1999*). It is therefore expected that cells minimize the amount of ribosomes they produce, making only the ribosomes required for protein translation in the given condition.

The coordination of ribosomal protein expression with cell growth rate is of a particular interest. Ribosomes drive protein translation and therefore cell growth. Increasing the amounts of ribosomes in cells could therefore increase growth rate. Yet, translation of ribosomal proteins competes with the production of other proteins and could therefore introduce limitations in other essential cellular processes, such as the production of metabolites needed for translation. Resource allocation models formalize this interplay by defining the optimal distribution of translation resources which maximizes growth rate. These models predict that growth rate is maximized when all expressed ribosomal proteins are fully employed in protein translation (*Bosdriesz et al., 2015*; *Dekel and Alon, 2005*; *Kafri et al., 2016a*; *Keren et al., 2013*; *Klumpp et al., 2013*; *Koch, 1988*; *Maaløe, 1979*; *Scott and Hwa, 2011*; *Scott et al., 2010*; *2014*; *Vind et al., 1993*). Indeed, producing excess ribosomal proteins that are not actively translating would compete with alternative cellular processes without contributing to protein production and cell growth.

The prevailing model therefore assumes that in growing cells, all ribosomes are constantly translating. However, recent studies suggest that while this model may hold in rapidly growing bacteria,

slow-growing bacteria do contain excess ribosomes that are not actively translating (*Dai et al., 2016*). Further, we recently reported that in rapidly growing budding yeast, protein translation is not universally limiting for protein production and growth rate (*Kafri et al., 2016b*). This raises the question of whether rapidly growing cells do in fact employ all expressed ribosomes at full capacity, as predicted by resource allocation models.

Consistent with its major role in driving cell growth, the ribosome content of bacteria growing in different environments increases linearly with growth rate, independently of the specifics of each nutrient (*Brauer et al., 2008*; *Bremer and Ehrenberg, 1995*; *Nomura et al., 1984*; *Schaechter et al., 1958*; *Scott et al., 2010*; *Waldron and Lacroute, 1975*; *Warner, 1999*; *Zaslaver et al., 2009*). This classical growth law was derived by measuring RNA content, which well approximates the amounts of ribosomes. In this work, we wished to extend these studies in three directions. First, we focus on budding yeast, asking whether the bacterial growth law connecting ribosome content and growth rate is conserved in this model eukaryote. Second, using proteomic profiling, we directly measure ribosomal protein levels and compare their abundance with that of other protein groups. Finally, we consider non-steady state conditions and in particular situations in which cells face increased translation demands, asking whether and how such conditions impact the relation between ribosome content and growth rate.

By analyzing the proteome composition of cells growing in a wide range of conditions, we show that the proteome fraction coding for ribosomal proteins scales linearly with growth rate, in a manner that is similar to that described in bacteria. Quantitative analysis of this scaling relation, coupled with polysome profiling, suggests that at each growth rate, a constant $\approx 8\%$ fraction of the entire proteome encodes for an excess of ribosomes that are not actively translating. Accordingly, in rapidly growing cells, $\approx 25\%$ of ribosomal proteins produced are not employed in translation at any given time and this inactive fraction significantly increases as the growth rate decreases. We provide evidence that this excess of ribosomes may be employed when cells are subjected to an unexpected increase in translation demands. Finally, we suggest that steadily growing cells prepare for fluctuating conditions by producing excess ribosomes. This ribosomal excess enables a faster response to upshift conditions, but comes at the expense of limiting the cells' instantaneous growth rate.

## Results

### The proteomic composition of budding yeast cells growing in different conditions

We employed mass-spectrometry analysis to define the proteomic composition of *S. cerevisiae* cells growing in three conditions: standard media (SC), media low in nitrogen (Low N) and media low in phosphate (Low Pi). This data was combined with published proteomic datasets of budding yeast growing on 12 different carbon sources (*Paulo et al., 2015*; *2016*). To avoid method-specific biases, all profiles were calibrated with an external data reference defining absolute protein levels (*Wang et al., 2012*).

Examining the Pearson correlation between the different profiles (*Figure 1A*), we observed that profiles were classified into two dominating clusters, depending on whether cells grew in a fermentative or respiratory mode. Correlations between profiles within the same cluster were $\approx 0.7$–$0.9$, while correlations between profiles assigned to different clusters were lower but still substantial (0.3–0.6). The data further demonstrated the expected induction of condition-specific proteins (e.g. activation of the phosphate and nitrogen starvation pathways), as well as differential expression of proteins involved in translation and stress response (*Figure 1B*).

Growth rate is a major determinant of proteome composition. Division times of cells in the 15 conditions we examined varied between 1.5 to 6.5 hr. To obtain a general overview of the effects of growth rate on the proteome composition, we classified proteins into eleven groups of related functions, which together covered $\approx 80\%$ of the proteome (*Supplementary file 1*), and examined how the relative abundance of each group changes with growth rate. In standard media, the proteome was dominated by translation-related factors ($\approx 40\%$) and glycolytic proteins ($\approx 15\%$) (*Figure 1C* and *Figure 1—figure supplement 1A*). The fraction of glycolytic proteins remained largely invariant between conditions, while the translation-related fraction decreased with growth rate, reaching $\approx 15\%$ in slow-growing cells. This decrease was accompanied by an increased abundance of

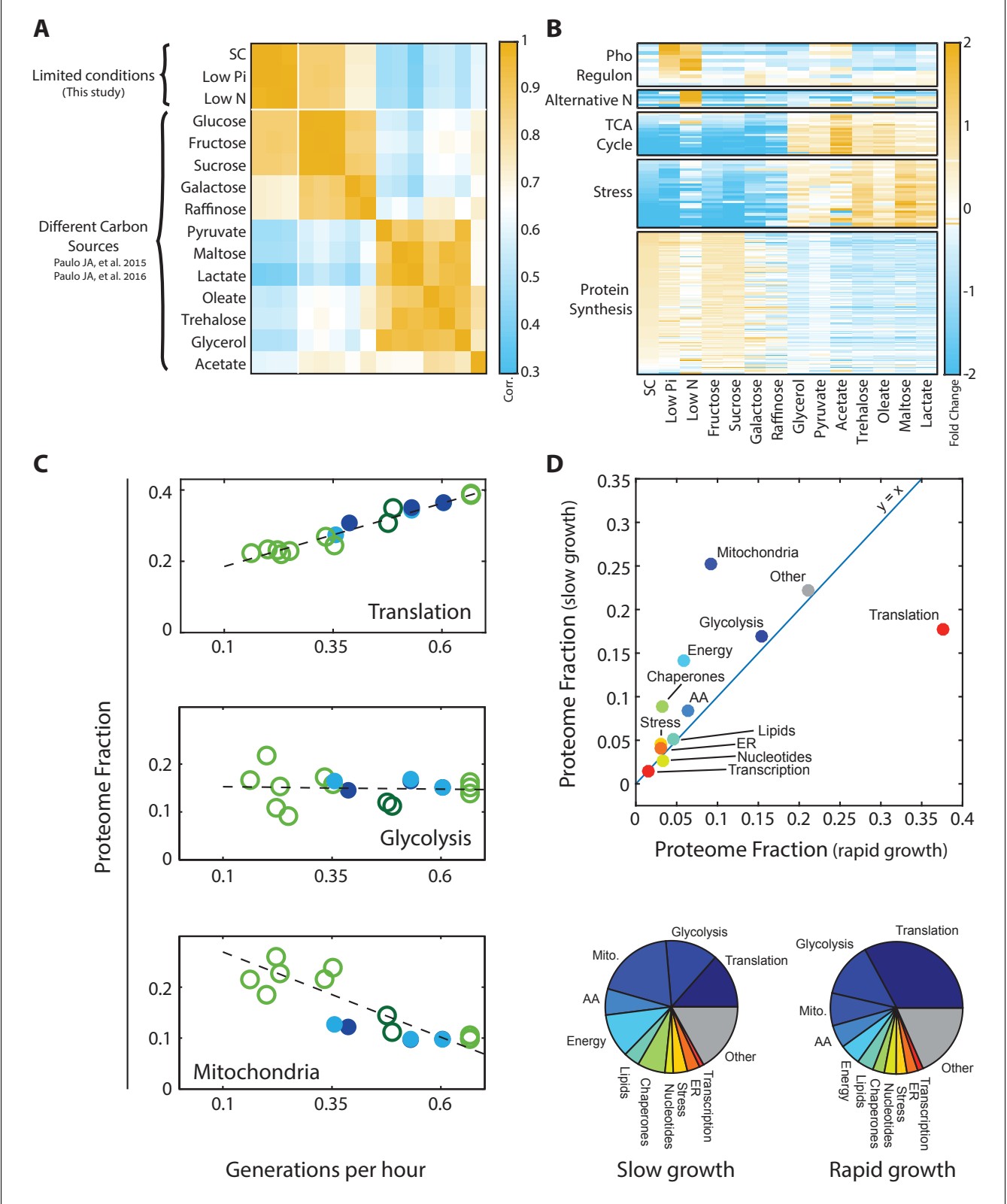

**Figure 1.** Proteomic analysis of budding yeast grown in different conditions. (**A**) Proteome profiles in our dataset clusters into two main groups on their fermentative or respiratory growth mode: Shown is the Pearson correlation matrix between proteome compositions in the indicated conditions. (**B**) Condition-dependent regulation of protein expression: The expression of each protein in each condition was normalized by its mean expression over all conditions. Shown is the (Log$_2$) protein expression of proteins in the indicated groups. See **Supplementary file 2** for protein names. (**C**) Expression

*Figure 1 continued on next page*

*Figure 1 continued*

of translation genes decreases in slow-growing cells: Proteins were classified into eleven groups by function, which together included ≈80% of the proteome. For each condition, we calculated the fraction of the proteome coding for each of these eleven groups. Shown here are the proteome fractions of groups composed of proteins involved in translation, glycolysis or mitochondrial function, plotted as a function of Generations per Hour (generation time$^{-1}$ = μ/ln(2)). Additional protein groups are shown in *Figure 1—figure supplement 1A*. Filled circles correspond to data obtained in this work while empty circles are data from *Paulo et al. (2015)*, *(2016)*, as specified in *Figure 2A*. The proteins assigned to each protein group are specified in *Supplementary file 1*. Dashed lines are the data's linear fits. Note that the translation group slope (y = 0.36x + 0.15) is almost identical to the ribosomal group slope (*Figure 2A*). (D) The overall proteome composition in fast vs. slow growing cells: The fraction of proteome encoding for each protein group was compared between the fast and slow growth condition in our dataset. In the upper panel, the proteome fraction encoding each specified protein group is plotted as a function of this group in the slow growing condition. Fast growth (0.67 gen/hr) corresponds to standard (SC) conditions, while slow growth (0.15 gen/hr) is extrapolated from the linear fit of the abundance vs. growth rate relation, shown in *Figure 1C* and *Figure 1—figure supplement 1A*. In the bottom panel, the same data is plotted as pie-charts.

DOI: https://doi.org/10.7554/eLife.28034.002

The following source data and figure supplement are available for figure 1:

**Source data 1.** DAmP and WT LC-MS/MS proteomic data as described in Materials and methods.

DOI: https://doi.org/10.7554/eLife.28034.004

**Figure supplement 1.** Proteome composition.

DOI: https://doi.org/10.7554/eLife.28034.003

condition-specific proteins, which in our dataset were mostly respiration-related. Mitochondrial proteins, for example, double in quantity to ≈20% of the proteome in the slow growing cells, in accordance with the increased reliance of these cells on respiration (*Figure 1C*). We summarize the changes in the proteomic fraction devoted to the different gene groups in slow versus fast growing conditions by plotting the abundance of each group in the slowest growth condition, against its abundance in the fast growth conditions (*Figure 1D*, top). This is also illustrated by a pie-chart summarizing the proteome composition in those two conditions (*Figure 1D*, bottom).

## The proteome fraction encoding ribosomal proteins scales linearly with cell growth rate

We next focused more specifically on the relation between the expression of ribosomal proteins and the rate of cell growth. We defined the ribosomal fraction of the proteome by clustering together all proteins annotated as subunits of the ribosome (*Supplementary file 1*). The proteome fraction coding for this group increased linearly with growth rate, irrespective of the specific media, from ≈8% in the slowest growing cells to ≈30% in rapidly growing cells (*Figure 2A*). Therefore, budding yeast show a similar growth law to that previously described in bacteria.

Cells may adjust their ribosome content by modifying protein translation, protein degradation or mRNA levels. To distinguish between these possibilities, we examined the transcription profiles of cells growing in the different conditions used for the proteome profiling. Notably, plotting the fraction of mRNA transcripts that code for ribosomal proteins as a function of cell growth rate showed the same quantitative scaling as observed at the level of the proteome (*Figure 2B* top). Thus, regulation of the ribosomal protein fraction with growth rate depends almost exclusively on mRNA transcription.

Next, we asked whether the scaling between ribosome content and growth rate is strain-specific and regulated by changing growth conditions, or whether it also explains differences in growth rate between different strains growing in the same condition. We recently reported that budding yeast strains show a large strain-to-strain variability in growth rate when growing on pentose as the sole carbon source (*Tamari et al., 2014*; *2016*). Examining the transcription profiles of these strains, we find that the ribosomal fraction follows growth rate with the same qualitative scaling as observed when comparing growth rates in different conditions (*Figure 2B*, top). The general scaling relationship between ribosome content and growth rate is therefore conserved not only when comparing the same strain across conditions, but also when comparing different strains growing in the same condition.

We next asked whether the scaling of ribosome content and growth rate is also maintained following genetic perturbations that affect cell growth. To this end, we used a large dataset describing the gene expression profiles and growth rates of most viable yeast deletion mutants

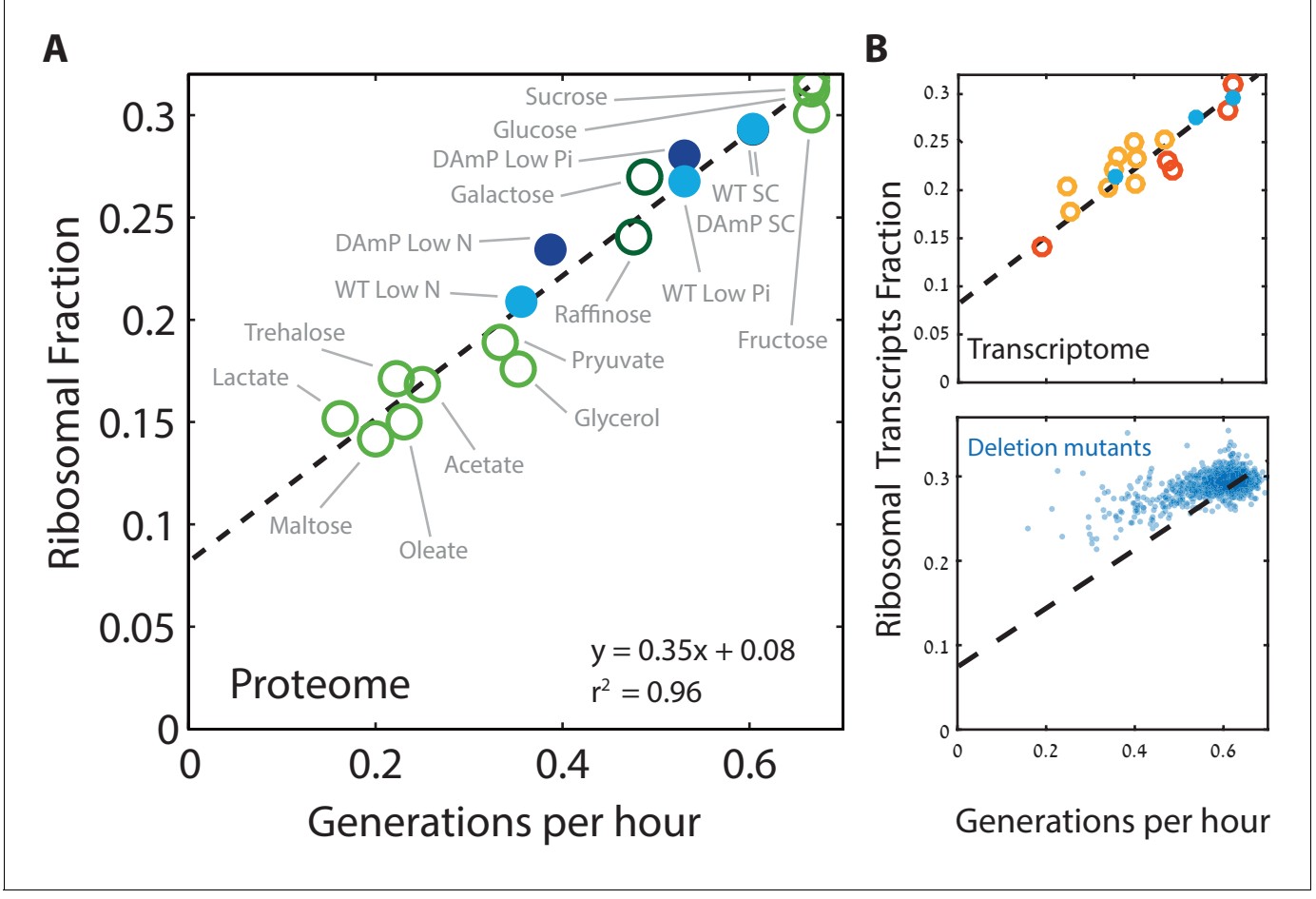

**Figure 2.** Ribosome content scales linearly with cell growth rate. (**A**) Shown is the fraction of the proteome coding for the ribosomal proteins in each condition as a function of cell growth rate. Note that here and henceforth, growth rate is shown in units of generation per hour (Generation Time$^{-1}$) which is related to the specific growth rate μ by a scaling factor, $μ = GT^{-1}*ln(2)$. The slope of this scaling curve is = 0.35×60 min=21 min so Δr/Δμ = 21/ln(2)[min]. For a broader definition of the translational group, see *Figure 1C* top panel and *Supplementary file 1* for gene names. Our data (filled circles) is generated from three biological repeats. (**B**) The transcriptome fraction coding ribosomal proteins scales with growth rate: Shown on top is the fraction of the ribosomal protein transcripts from the full transcriptome as a function of cell growth rate. Conditions are specified in *Figure 2— figure supplement 1A*. Dotted black line is the linear fit from (**A**). Note the high agreement between the transcriptomic data and the proteomic linear fit, implying that on average, the translation-to-degradation of ribosomal proteins is the same as that of other proteins (primarily highly abundant ones). The bottom figure shows that this scaling is lost when comparing growth-affecting mutations. Mutant data was taken from *Kemmeren et al. (2014)* and *O'Duibhir et al. (2014)*.

DOI: https://doi.org/10.7554/eLife.28034.005

The following figure supplement is available for figure 2:

**Figure supplement 1.** The proteome profiles of budding yeast cells growing in different conditions.

DOI: https://doi.org/10.7554/eLife.28034.006

(*Kemmeren et al., 2014*; *O'Duibhir et al., 2014*). Notably, in this case, the scaling between ribosome content and growth rate was largely lost (*Figure 2B*, bottom); in fact, most mutants that reduced growth rate had a minor effect on the expression of genes coding for ribosomal proteins. As a result, growth-affecting mutations expressed more ribosomal protein coding genes than expected given their growth rate. We observed a similar deviation from the scaling curve in cells forced to express a large amount of unstable mRNA (*Figure 2—figure supplement 1B–C*): here as well, growth rate decreased but expression of ribosomal proteins remained largely unaffected. Therefore, while the scaling of ribosome content with growth rate is well conserved in wild-type strains growing in different conditions, it is not maintained upon growth-affecting genetic mutations, primarily because most growth-affecting mutants do not lead to a comparable reduction in the

ribosome content. This implies that the tight correlation between ribosome content and growth rate results from environmental signaling, rather than direct causal relationships (*Levy et al., 2007*).

## Quantitative interpretation of the scaling between ribosome content and growth rate

The scaling relation between ribosome content, $r$, and the specific growth rate, $\mu$, is typically interpreted based on the general relationship between protein translation and growth rate during balanced growth. In balanced growth, the cellular protein content doubles at each cell division. Since proteins are translated by active ribosomes, this leads to the relationship: $\mu = \gamma\, r_a$, where $\gamma$ is the rate of ribosome translation and $r_a$ the proteome fraction encoding for actively translating ribosomal proteins (*Kafri et al., 2016a*; *Maaløe, 1979*; *Scott et al., 2010*). Note that $\gamma^{-1}$ is the time required for one ribosome to duplicate itself, defining a limit on the minimal cell cycle time.

Written differently, this relationship connects the two parameters we measure: the specific growth rate $\mu$, and the proteome fraction encoding for the total (translating and non-translating) ribosomal proteins, $r$:

$$r = \gamma^{-1}\, \mu + r_0 \tag{1}$$

Here, $r_0 = r - r_a$, is the proteome fraction encoding for residual ribosomal proteins that are not actively translating. In principle, both parameters of this theoretical relation, $\gamma$ and $r_0$, could change with nutrient conditions. The measured scaling curve, *Figure 2A*, however, implies that these parameters are tightly coordinated to maintain the linear relationship between ribosome content and growth rate across the different conditions.

We can estimate $\gamma$ using measured parameters: the translation elongation rate (10.5 aa/sec) (*Waldron et al., 1977*) and the number of amino-acids composing the budding yeast ribosome (12,485 A.A). By this estimation, $\gamma^{-1} = 20$ min. This predicted value is $\approx 30\%$ smaller than our measured slope $\Delta r / \Delta \mu = 21/\ln(2)$ min, (*Figure 2A–B*). Of note, the same quantitative difference between the estimated and measured slope was seen in bacteria, where it was interpreted as inaccuracy in estimated $\gamma$, stemming from the need to include additional translation factors that function within the active ribosomal unit (*Scott et al., 2010*). Alternatively, this discrepancy between the estimated and measured slope may reflect inaccuracies of the simplified theory. For example, any time-delay between production of ribosomal protein and their contribution to the actively translating pool will increase the predicted $\gamma^{-1}$. In addition, the measured slope would be precisely explained if ribosome production in each cell was effectively linear rather than exponential. In this case, the relation between the specific growth rate $\mu$ and the division time $T_d$ is $\mu = 1/T_d$ rather than $\mu = \ln(2)/T_d$.

## A constant fraction $r_0 \approx 8\%$ of the proteome encodes for non-translating ribosomal proteins

In our analysis above, we distinguished the fraction of ribosomal proteins that contribute to cell growth ($\mu = \gamma\, r_a$) from the residual inactive, non-translating part ($r_0$). The fact that inactive $r_0$ exists is most easily appreciated at the regime of very slow, or no-growth. In this regime, ribosomal proteins still account for $r_0 \approx 8\%$ of the proteome (*Figure 2A–B*). As protein translation in arrested cells is dramatically reduced, the majority of these ribosomes remain inactive. Indeed, arrested cells were shown to maintain a pool of inactive ribosomes (*Dai et al., 2016*; *van den Elzen et al., 2014*).

By extension, residual inactive ribosomes may also be present in proliferating cells. Using the simplified model above, *Equation (1)*, the fraction of this inactive pool $r_0$ can be estimated from the measured scaling curve, provided that we know the ribosome elongation rates $\gamma$. If elongation rate $\gamma$ is constant and $\gamma$ is independent of growth rate, then the residual fraction also remains a constant $r_0 = 8\%$ at all growth rate. By contrast, if elongation rate changes with growth rate, $\gamma = a^{-1}\mu$, where $a$ is a fixed growth-rate independent constant, then the residual fraction should also increase with the growth rate, $r_0 = r - a$. Similarly, the $r_0/r$, fraction of inactive ribosomes out of the total ribosome pool, will increase with decreasing growth rate if elongation rate is constant (*Figure 3A*), but will decrease with decreasing growth rate if elongation rate changes.

Previous measurements of elongation rates in different media gave conflicting results. In one report, measured elongation rates in glucose and in media lacking nitrogen remain the same despite a two-fold reduction in growth rate (*Waldron et al., 1977*). By contrast, a second report suggested

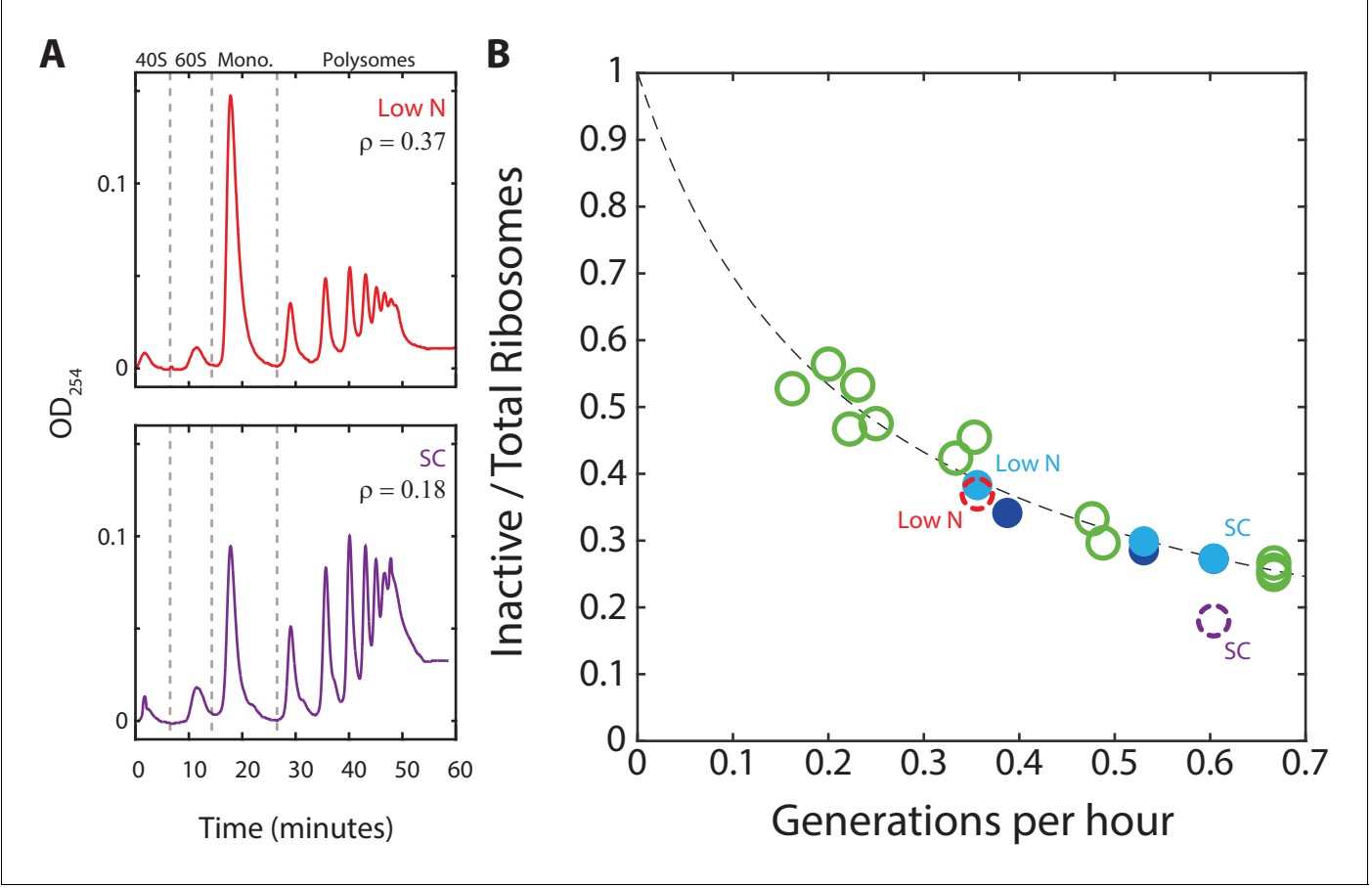

**Figure 3.** A substantial fraction of ribosomes is not actively translating at a given time. (**A**) Estimating the fraction of inactive ribosomes using polysomal profiling: Cells were grown in the indicated conditions and their ribosomal content was analyzed on sucrose gradients as an indication for translational activity. Representative profiles of raw data are shown. The fraction of inactive ribosomes was estimated by the ratio (ρ) of monosomes (mRNAs bound by a single ribosome), the 40S and 60S, to the total ribosome density and is plotted in (**B**) in corresponding colors (dashed circles). (**B**) *The fraction on inactive ribosomes increases with decreasing growth rate:* Shown is the estimated fraction of inactive ribosomes ($Y = 0.08/(0.35X + 0.08)$) for each condition, as a function of cell generations per hour. Conditions as specified in *Figure 2A*. Dashed circles represent the inactive/total ribosomes ratio (ρ) calculated in (**A**) from three independent experiments.

DOI: https://doi.org/10.7554/eLife.28034.007

that elongation rates decrease by ≈40% in media lacking acetate (*Bonven and Gulløv, 1979*). These measurements used the same experimental setup but differed in the details of the analysis (See Material and methods).

To more directly examine the fraction of inactive ribosomes in growing cells, we performed polysomal profiling. In this method, the optical density of RNAs is quantified, with polysomes consisting of mRNA bound by multiple ribosomes. Most actively translating ribosomes are found in the polysomes while monosomes (mRNA with a single ribosome) are thought to contain primarily inactive ribosomes (*Aspden et al., 2014*; *Castelli et al., 2011*; *Chassé et al., 2017*; *Kapp et al., 2004*; *Kelen et al., 2009*; *Liu and Qian, 2016*; *Warner et al., 1963*). Therefore, using the monosome to polysome ratio, we can estimate the fraction of inactive ribosomes out of the total ribosome pool and compare it to the predicted behavior, as above.

We examined the polysomal profiles in cells growing in standard medium (SC), in cells growing in low phosphate and low-nitrogen media. A substantial inactive fraction (monosomes) was observed in the fast growing cells (ρ = 0.18), which further increased in the slow-growing cells (*Figure 3A*). Notably, the relative fraction of monosomes increased with decreasing growth rate. Further, this fraction was in quantitative agreement with the predicted fraction of inactive ribosomes assuming a constant residual $r_0$ ≈ 8% (*Figure 3B*). We conclude that during steady growth, cells devote a constant, ≈8%

of their proteome to producing ribosomes that are not actively translating. Therefore, the fraction of ribosomes that are not actively translating ranges from $\approx 25\%$ in rapidly growing cells ($r_0 \approx 8\%$ relative to the total ribosomal fraction $r \approx 30\%$) and further increases with decreasing growth rate (*Figure 3B*).

## The scaling between ribosome content and growth rate changes when cells prepare to enter stationary phase

Cells may use the excess ribosomes to accommodate an increase in translation demands. Entering stationary phase represents such a scenario (*Ju and Warner, 1994*), as the reduced availability of nutrients requires an increased production of glycolytic enzymes (*Figure 1—figure supplement 1A*, right column). To examine whether the fraction of non-translating ribosomes ($r_0$) decreases during this transition, we analyzed published proteomic profiles of batch-grown cultures entering stationary phase (*Murphy et al., 2015*). Indeed, ribosomal content decreased $\approx 2.5$ generations (four hours) before growth rate dropped (*Figure 4A*, *Figure 4—figure supplement 1C*). This could be interpreted in two ways: a transient increase in translation elongation rate, or a more efficient use of available ribosomes, leading to lower $r_0$.

Similarly, transcription profiling of cells shifted to low phosphate medium showed an early decrease in ribosomal protein transcripts, ~5 hr before growth rate was reduced (*Figure 4—figure supplement 1A–B*). Therefore, during preparation for stationary phase, cells reduce their overall ribosomal content but maintain a stable growth rate. Again, this could be interpreted in two ways: either cells transiently increase the translation elongation rate, or they make a more efficient use of the already available ribosomes by decreasing the non-translating fraction $r_0$.

## The scaling between ribosome content and growth rate changes during nutrient upshift

Translation demands increase when cells undergo nutrient upshifts. Under such conditions, nutrient limitation is lifted, and cells can resume fast growth practically immediately, provided that sufficient ribosomes are available to enable faster protein production. We examined the kinetics by which cells resume fast growth upon nutrient upshift by reanalyzing data we previously acquired. First, we found that continuous cell cultures growing in a phosphate-limited chemostat responded to a spike of high phosphate within minutes (*Figure 4B*). Second, using microfluidic-coupled microscopy, we followed individual cells transferred from galactose to glucose and observed that cells increase their growth rate from 0.36 gen/hr to 0.54 gen/hr within minutes of the transfer (*Figure 4C*). Therefore, cells increase their growth rate and thus protein production rates within a short time that appears insufficient for synthesizing new ribosomes. This is consistent with either a transient increase in translation elongation rate, or a more efficient use of available ribosomes through a decrease in the non-translating fraction $r_0$.

To more directly correlate the ribosomal protein content and growth rate during nutrient upshift, we profiled the proteome of cells transferred from early stationary phase ($OD_{600} \approx 6$) to fresh media (SC). Cells resumed rapid growth of 0.55 gen/hr following a $\approx 40$ min lag. Proteome profiling revealed that ribosome content did not change during the lag time, but began to increase only after cells had attained fast growth (*Figure 4D*). Since active translation in stationary cells is minimal, we can conclude that in this case, cells necessarily made use of the residual ribosomal fraction $r_0$ for initiating growth. In fact, it appears that to account for the observed growth rate during this initial growth phase, cells need to utilize the vast majority of expressed ribosomes.

## The excess of non-translating ribosomal proteins is decreased when cells are forced to express unneeded proteins

The linear scaling between ribosome content and growth rate is adjusted when cells face increased translation demands. This implies a transient increase in translation elongation rate or employment of the residual ribosomal fraction $r_0$. To better understand these effects, we asked whether increasing translation demands during logarithmic growth will also alter the scaling relation between ribosome content and growth rate. To this end, we engineered cells to constitutively produce high mCherry protein amounts using a system we previously described (*Kafri et al., 2016b*). In this

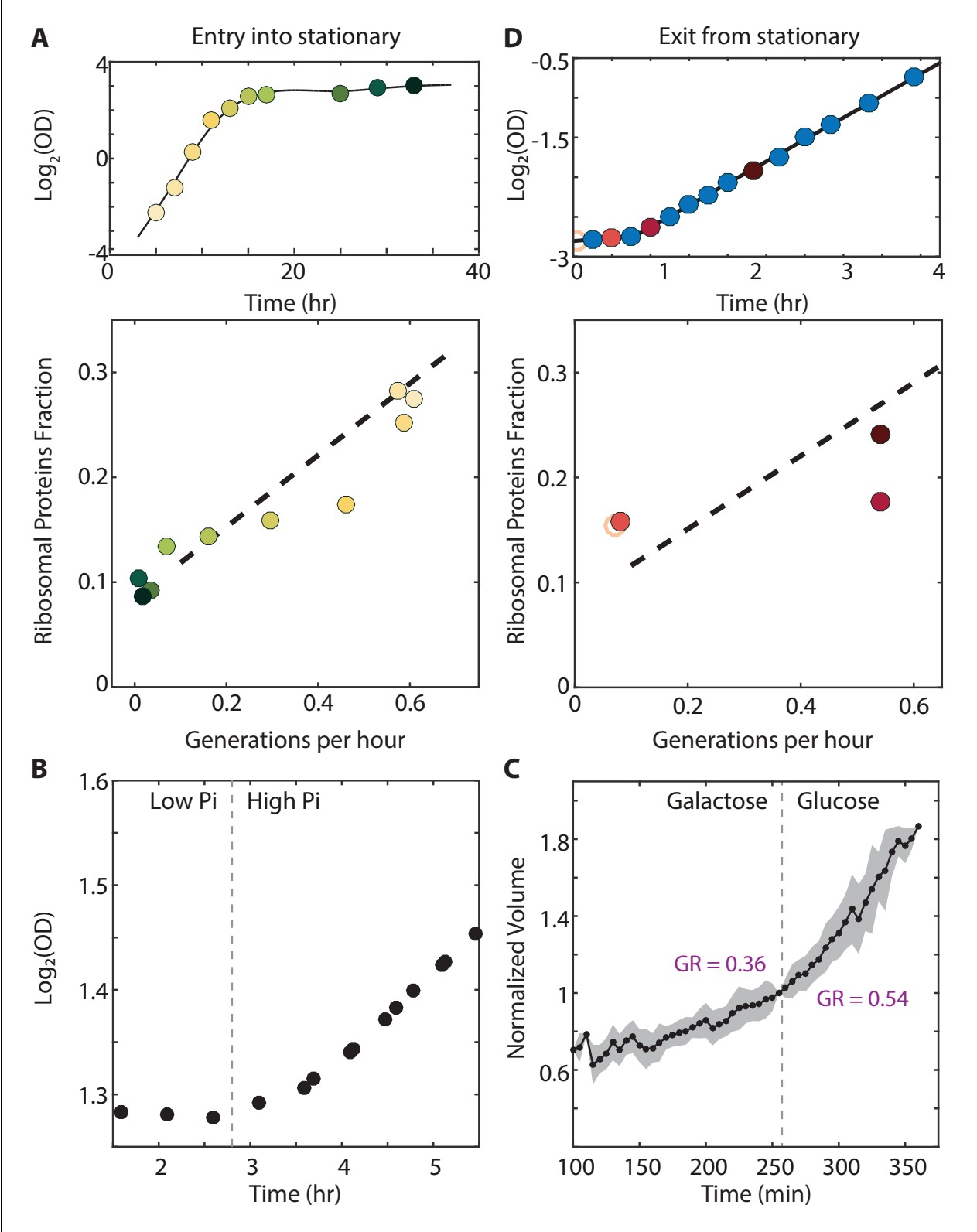

**Figure 4.** The scaling between ribosome content and growth rate changes when balanced growth is perturbed. (**A**) The scaling between ribosome content and growth rate changes when cells prepare to enter stationary phase: Cells grown in batch culture were followed as they increase in density (top panel), and were subjected to proteome profiling at the indicated times. The bottom panel shows the fraction of proteome coding for ribosomal proteins along this time course. Data taken from **Murphy et al. (2015)**. Color gradient from light to dark represents increasing time. (**B**) Cells growing

*Figure 4 continued on next page*

*Figure 4 continued*

in a phosphate-limited chemostat increase their growth rate immediately upon phosphate addition: Continuous cultures were grown to steady state in a phosphate-limited chemostat at a dilution rate of 4.5 hr/gen. Media containing high phosphate was then injected into the growth chamber (dotted line). Shown is the density (OD) of the culture before and after phosphate injection. (C) Cells increase their volume growth immediately upon transfer to a preferred carbon source: Cells were imaged using microfluidics-coupled live-cell microscopy, while their carbon source was changed from galactose to glucose (dotted line). Shown is the cell volume before and after the media change, averaged over 50 cells. Specific growth rate was calculated 25 min before and after the upshift. Error bars represent SEM. (D) Cells exiting stationary phase resume rapid growth before increasing the ribosome content: Cells were grown in SC until saturation ($OD_{600} \approx 6$) and diluted back into fresh media at time 0 (empty circle). OD measurements were taken every 15–20 min as shown in the top panel and samples for proteomic analysis were taken as indicated. The ribosomal fraction, as calculated from the proteomic data, is shown in the bottom panel as a function of cell growth rate. Each point is the median of three biological repeats.

DOI: https://doi.org/10.7554/eLife.28034.008

The following figure supplement is available for figure 4:

**Figure supplement 1.** The scaling between ribosome content and growth rate changes when balanced growth is perturbed.

DOI: https://doi.org/10.7554/eLife.28034.009

analysis, a library of strains expressing increasing amounts of mCherry proteins was generated, with mCherry levels reaching $\approx 25\%$ of the total proteome in the highest burden cells.

We measured the growth rate and profiled the proteome for cells with different levels of mCherry burden in three conditions: standard medium (SC), medium containing low-phosphate and medium containing low nitrogen (*Figure 5—figure supplement 1C*). As expected, in all three conditions, growth rate and the ribosomal fraction $r$ decreased with increasing mCherry levels (*Figure 5A*). For each level of burden, we plotted the ribosome content as a function of the cell's growth rate in the three growth conditions examined (*Figure 5A*, *Figure 5—figure supplement 1A*). Burden cells showed a scaling relation that was highly similar to that observed in wild-type cells: when compared across conditions, ribosome content decreased linearly with growth rate. Notably, the slope of this scaling curve was independent of the level of the burden, suggesting that ribosome translation rates $\gamma$ remained invariant to the burden.

In contrast, the burden did affect the residual $r_0$, the predicted proteome fraction coding for non-translating ribosomes. This can be easily appreciated from the zero-growth limit. In this limit, ribosome content in high burden cells decreased from its wild-type value ($r_0 = 8.1\%$) to a lower value, $r_0 = 5.5\%$ (*Figure 5—figure supplement 1B*). Therefore, the burdened cells employ a higher fraction of their ribosomes, leaving a smaller pool of inactive ribosomes.

## The smaller pool of excess ribosomes in burden cells explains their delayed recovery from starvation

Burdened cells exit starvation with a delay (*Kafri et al., 2016b*; *Shachrai et al., 2010*). As we showed above, during this exit, cells employ a large fraction of the excess ribosomal proteins $r_0$ to initiate protein translation and cell growth. We therefore reasoned that the delayed recovery of the burdened cells may be due to the smaller fraction of the residual ribosomes, $r_0$, they express.

To examine this prediction, we quantified the delay introduced by the burden when cells recovered from starvation (*Figure 5B*; data from *Kafri et al. (2016b)*. As expected, this delay increases linearly with the mCherry burden. Notably, the differences in recovery times were quantitatively explained by the difference in $r_0$; for example, the recovery time of eight-copy burden cells was prolonged by $\approx 15\%$, consistent with the corresponding $\approx 15\%$ decrease in the measured $r_0$.

## The fraction of ribosomes actively translating is tuned by growth conditions, but remains largely invariant to cell growth rate

A central implication of the scaling relation between ribosome content and growth rate (*Figures 2A* and *5A*) is that when growth conditions change, cells tune not only the fraction of the proteome encoding for ribosomal proteins ($r$), but also the ratio between active and inactive ribosomes ($r_a/r_0$). As a direct consequence of the scaling law, when comparing cells that grow in different conditions, the predicted ratio $r_a/r_0$ increases linearly with growth rate (*Figure 6A*). However, comparing between wild-type and burden cells, we noted that this ratio is invariant to the burden and depends mainly on the growth condition (*Figure 6A*). Thus, although burden cells grow more slowly and express a smaller ribosomal protein fraction compared to wild-type cells, the fraction which is

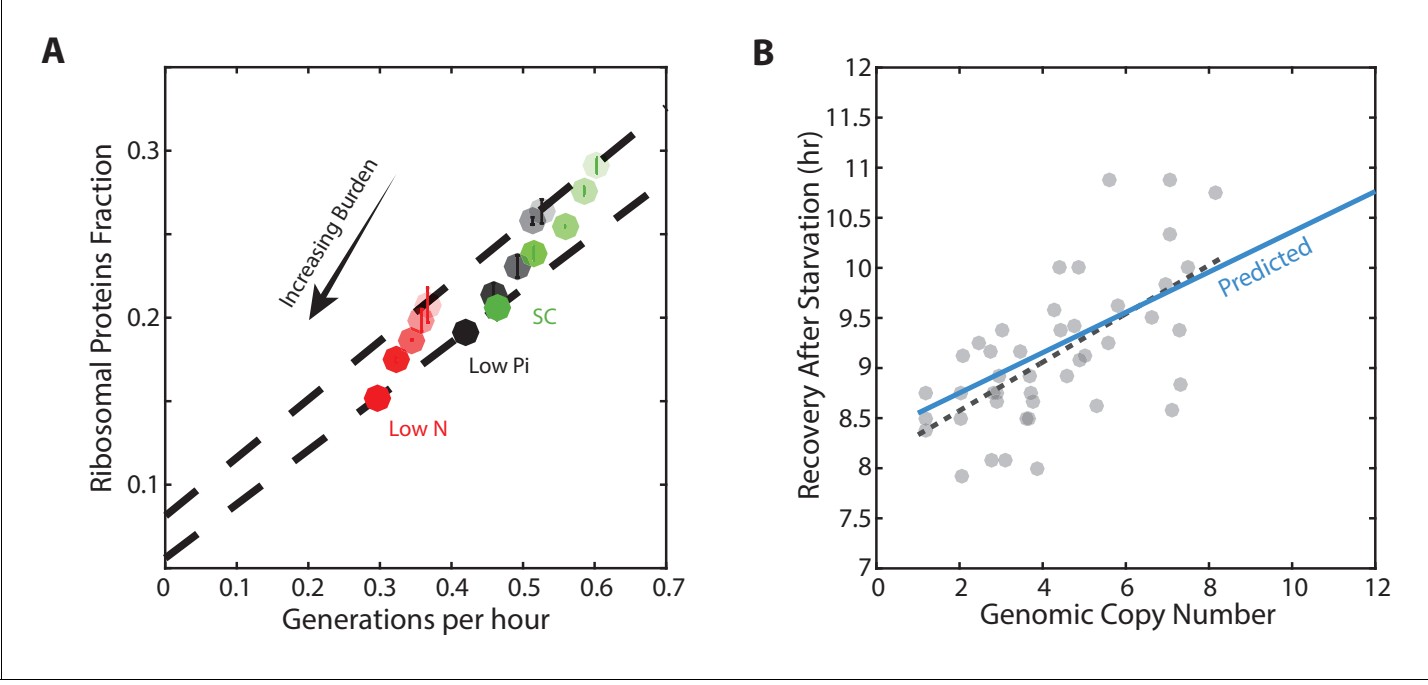

**Figure 5.** Forcing unneeded protein production reduces the pool of free ribosomes. (**A**) Scaling of ribosomal proteins with growth rate in protein-burdened strains: Five strains expressing increasing amounts of mCherry proteins were generated and their proteome profiles and growth rates were measured in the three indicated conditions. Shown is the proteome fraction coding for ribosomal proteins in each strain and in each condition as a function of the cell growth rate. Different strains are indicated by the different shades of colors, with increased burden corresponding to a darker circle. Colors indicate the specific condition used. The two dashed lines correspond to the scaling curves defined by the no-burden and most highly-burdened strains: The top line is the same as in *Figure 2A*, while the bottom line describes the ribosome content of the highest burden as a function of its growth rate in the different conditions. Error bars represent the standard deviation around the median between three biological repeats. (**B**) The reduced pool of inactive ribosome in the burden cells quantitatively accounts for their delayed exit from starvation: Cells expressing different amounts of mCherry proteins were grown to saturation. The cells were kept in stationary for a week before diluting them back to fresh media. Recovery times (Y axis) were defined by the time at which cells' optical density increased by 50%. Solid blue line represents the predicted recovery time based on the fold-reduction in free ribosomes, dashed black line is the data's linear fit. Data is re-plotted and analyzed from *Kafri et al. (2016b)*.

DOI: https://doi.org/10.7554/eLife.28034.010

The following figure supplement is available for figure 5:

**Figure supplement 1.** Cells forced to produce unneeded proteins have a smaller pool of excess ribosomes.

DOI: https://doi.org/10.7554/eLife.28034.011

employed in translation is predicted to remain the same as in wild-type cells, depending only on the growth conditions. To verify this prediction, we used polysomal profiling of wild-type and burdened strains growing in different conditions. As predicted, the ratio of polysomes to monosome remains largely invariant between wild-type and burden cells growing in the same condition, despite their different growth rate (*Figure 6B*).

Taken together, we propose that the tuning of ribosome content and of the fraction of ribosomes that are actively translating depend primarily on signaling from the environment with little contribution of internal growth-rate dependent feedback (*Figure 7*). Thus, although both parameters show a tight correlation with growth rate, this correlation is not direct, but rather results from evolutionarily tuned signaling.

## Discussion

Ribosome production and function are the major resource-consuming processes in cells and its tight control in response to a variety of signals is well appreciated (*Nomura et al., 1984*; *Reuveni et al., 2017*; *Schaechter et al., 1958*; *Schimmel, 1993*; *Scott et al., 2010*; *Warner, 1999*). Our study

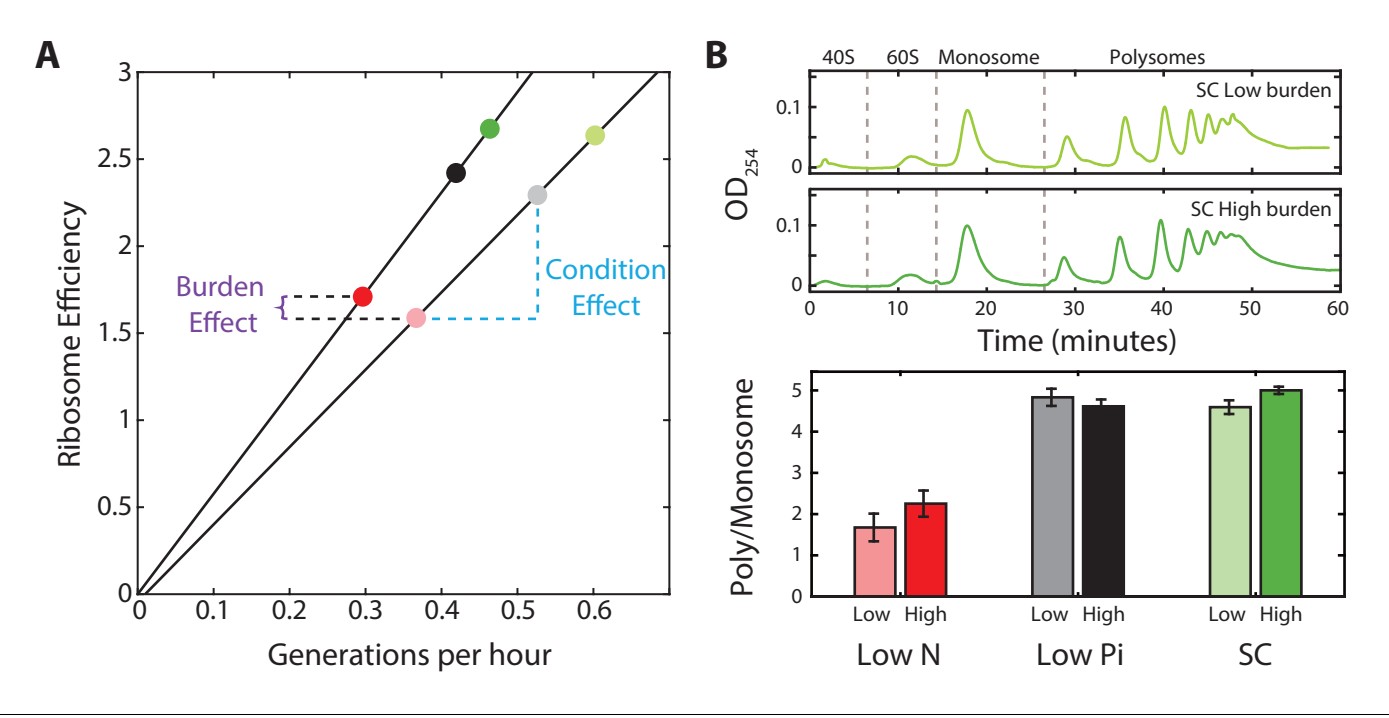

**Figure 6.** The ratio of active to inactive ribosomes remains invariant to protein burden. (**A**) The ratio of active to inactive ribosome predicted by proteomic data: Shown is the ratio of active to inactive ribosomes $r_a/r_0$, as predicted from our analysis of proteomic data as a function of growth rate in wild-type and high-burden cells. Note that while this ratio decreases with growth rate when comparing the same strain across different conditions, it remains almost invariant to the burden when compared between two strains within the same condition. (**B**) The ratio of active to inactive ribosomes predicted by polysome profiling: Low and high burden cells were grown in the indicated conditions and their ribosomal content was analyzed on sucrose gradients as an indication for translational activity. Representative profiles of raw data are shown on top. The ratio between the polysomes (active transcripts with more than one ribosome bound) to detached small and large subunits (40S and 60S) together with the monosomes (mRNAs bound by a single ribosome) is shown in a bar graph on bottom. SEM error bars are from three biological independent repeats.
DOI: https://doi.org/10.7554/eLife.28034.012

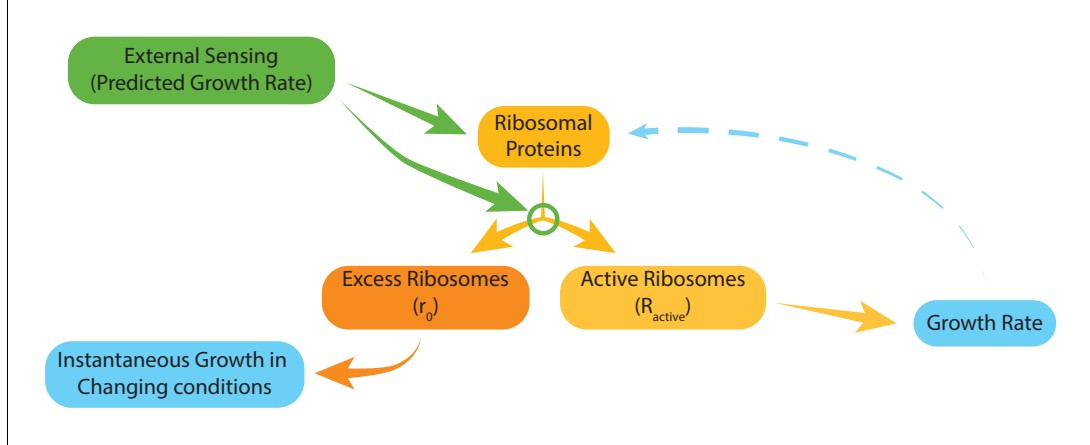

**Figure 7.** Model for ribosome allocation. Cells tune their ribosome content and ribosome efficiency based on signaling from the environment. Evolutionary tuning of this signaling results in a precise scaling of ribosome content with growth rate during logarithmic growth. Growth-rate dependent feedbacks play a minor role in the tuning of ribosome content or efficiency.
DOI: https://doi.org/10.7554/eLife.28034.013

quantified a specific aspect of this control that is particularly relevant for understanding the overall regulation of cellular physiology: how the proteomic fraction coding for ribosomal proteins is regulated with growth rate. This question was extensively studied in theoretical models and experimentally in bacteria (*Brauer et al., 2008*; *Bremer and Ehrenberg, 1995*; *Nomura et al., 1984*; *Schaechter et al., 1958*; *Scott et al., 2010*; *Waldron and Lacroute, 1975*; *Warner, 1999*; *Zaslaver et al., 2009*), but received relatively little attention in eukaryotic models.

We find that also in budding yeast, the proteome fraction coding for ribosomal proteins scales linearly with the specific growth rate. The value of this slope is predicted by models of balanced growth to be precisely the time required for translation elongation of a full ribosome. The quantitative agreement of our measured slope and this predicted value depends on whether we assume exponential or linear production of ribosomes at each individual cell cycle time: a precise agreement is obtained if linear growth is assumed, while the values differ by $\approx 30\%$ ($\approx \ln(2)$) if growth is exponential. Notably, this was also the case in bacteria: the measured slope differed by $\approx 30\%$ form the predicted slope calculated by assuming exponential growth. Since growth is indeed thought to be exponential, the discrepancy between the measured and predicted slopes may be explained by inaccuracies of the simplified theory, e.g. failure to include possible delays between ribosome production and usage which may effectively shift growth dynamics closer to linear growth. Alternatively, studies in bacteria attributed this difference to the need for including additional proteins for production of functional ribosomal unit, which may change the predicted value of the slope (*Klumpp et al., 2013*; *Scott et al., 2010*).

We provided evidences that a constant fraction ($\approx 8\%$) of the budding yeast proteome encodes for ribosomal proteins that are not actively translating at a given time. Accordingly, even rapidly growing cells still maintain $\approx 25\%$ of their ribosomal proteins inactive. This result is surprising in light of the prevailing notion that cells tune their proteome composition in a way that ensures that all expressed ribosomal proteins are employed at full capacity (*Bosdriesz et al., 2015*; *Dekel and Alon, 2005*; *Kafri et al., 2016a*; *Keren et al., 2013*; *Klumpp et al., 2013*; *Koch, 1988*; *Maaløe, 1979*; *Scott and Hwa, 2011*; *Scott et al., 2010*; *2014*; *Vind et al., 1993*). Notably, while the scaling of ribosome with growth rate was maintained over a range of conditions, it was readily altered by different perturbations. These included genetic mutations, as well as transient perturbations that led to a sudden increase in translation demands. Further, forcing the cells to express unneeded proteins effectively reduced the residual fraction of unused ribosomes, leading to a more efficient ribosome usage. These results imply that the scaling curve we observed, and in particular the fraction of residual, inactive ribosomes, is not an inherent unavoidable consequence of cell physiology or the mechanism of ribosome production, but is rather a consequence of signaling which sets the ratio of active vs. inactive ribosomes as a function of growth conditions.

Why would cells express extra ribosomes more than needed for meeting translation demands? We suggest that growing cells produce extra ribosomes in order to prepare for fluctuating conditions. Consider a nutrient upshift during growth in poor media: when the nutrient limitation is lifted, cells can increase their growth rate, provided that sufficient ribosomes are available. If cells were to precisely tune their ribosomal content to express only the ribosomes needed for the slow growing conditions, their level would limit the speed by which rapid growth could be resumed upon nutrients supply. Previous studies described strategies by which cells prepare for changing conditions, including the activation of stress genes by moderate stresses (*Gasch et al., 2000*; *Guan et al., 2012*; *Levy et al., 2011*; *Mitchell et al., 2009*) and the transition of a subpopulation into persistence (*Balaban et al., 2004*; *Levy et al., 2012*; *Soll and Kraft, 1988*; *Yaakov et al., 2017*). Our study suggests that this requirement to be prepared for fluctuating conditions affects not only specific genes or pathways, but also the central cellular processes such as the coordination of ribosome production with growth rate.

## Materials and methods

### Media and strains

All budding yeast (*S. cerevisiae*) strains in this study were based on BY4742 (*Brachmann et al., 1998*) or Y8205 (*Hin Yan Tong and Boone, 2005*) laboratory strains. TDH3p-mCherry (protein fusion) was made by fusing the mCherry cassette to the end of the Tdh3 protein using a standard

PEG;LiAC;ssDNA transformation protocol (*Gietz and Woods, 2002*). The rest of the strains were constructed previously (*Kafri et al., 2016b*).

Strains were grown in SC medium or in SC medium depleted of a specific nutrient, as described in the main text. SC limiting media was prepared from YNB without the relevant nutrient (Low Phosphate medium - ForMedium, CYN0804, Low Nitrogen medium - BD 3101130). Phosphate depleted medium was prepared by adding phosphate in the form of $KH_2PO_4$ to a final concentration of 0.2 mM. The level of potassium was preserved by adding KCl (instead of $KH_2PO_4$). Nitrogen limiting medium was prepared from YNB without amino acids and ammonium sulfate (BD 3101130) by adding separately 50 μM of ammonium sulfate and the essential amino acids.

## Growth rates

Growth rates were calculated from the data reported previously (*Kafri et al., 2016b*). We further validated the growth rates in the various conditions by log-phase competition assays (*Figure 5—figure supplement 1A*), as described below.

## Log-phase competition

Cells were grown overnight to stationary phase in the relevant media. GFP and mCherry strains were then co-incubated in the specified media at 30°C. The initial OD was set to ≈0.05, and the WT initial frequency was ≈50% of the total population. The number of generations was calculated from the dilution factor. Frequencies of GFP versus mCherry cells were measured by flow cytometry. The cells were diluted every ≈8 hr. Experiments were done with WT strains expressing GFP vs. mCherry burdened cells. A linear fit of the $log_2$ for the WT frequency was used to calculate the slope for each competition. The relative fitness advantage is calculated from the slope divided by $log_2$. The slope of the strains' fitness advantage and their copy number were used to calculate the growth effect per burden copy.

## Protein measurements

### Sample preparation

All chemicals are from Sigma Aldrich, unless stated otherwise. Samples were subjected to in-solution tryptic digestion using a modified Filter Aided Sample Preparation protocol (FASP). Sodium dodecyl sulfate buffer (SDT) included: 4%(w/v) SDS, 100 mM Tris/HCl pH 7.6, 0.1M DTT. Urea buffer (UB): 8 M urea (Sigma, U5128) in 0.1 M Tris/HCl pH 8.0 and 50 mM Ammonium Bicarbonate. Cells were dissolved in 100 μL SDT buffer and lysed for 3 min at 95°C and then centrifuged at 16,000 RCF for 10 min. 100 ug total protein were mixed with 200 μL UB and loaded onto 30 kDa molecular weight cut-off filters and centrifuged. 200 μl of UB were added to the filter unit and centrifuged at 14,000 x g for 40 min. Alkylation using 100 μl IAA, 2 washed with Ammonium Bicarbonate. Trypsin was then added and samples incubated at 37°C overnight. Additional amount of trypsin was added and incubated for 4 hr at 37°C. Digested proteins were then spun down to a clean collecting tube, 50 ul NaCl 0.5M was added and spun down, acidified with trifluoroacetic acid, desalted using HBL Oasis, speed vac to dry and stored in −80°C until analysis.

### Liquid chromatography

ULC/MS grade solvents were used for all chromatographic steps. Each sample was loaded using split-less nano-Ultra Performance Liquid Chromatography (10 kpsi nanoAcquity; Waters, Milford, MA, USA). The mobile phase was: (A) H2O + 0.1% formic acid and (B) acetonitrile +0.1% formic acid. Desalting of the samples was performed online using a reversed-phase C18 trapping column (180 μm internal diameter, 20 mm length, 5 μm particle size; Waters). The peptides were then separated using a T3 HSS nano-column (75 μm internal diameter, 250 mm length, 1.8 μm particle size; Waters) at 0.35 μL/min. Peptides were eluted from the column into the mass spectrometer using the following gradient: 4% to 22%B in 145 min, 22% to 90%B in 20 min, maintained at 95% for 5 min and then back to initial conditions.

### Mass spectrometry

The nanoUPLC was coupled online through a nanoESI emitter (10 μm tip; New Objective; Woburn, MA, USA) to a quadrupole orbitrap mass spectrometer (Q Exactive Plus, Thermo Fisher Scientific,

Bremen, Germany) using a FlexIon nanospray apparatus (Proxeon). Data was acquired in DDA mode, using a Top20 method. MS1 resolution was set to 70,000 (at 400 m/z) and maximum injection time was set to 20msec. MS2 resolution was set to 17,500 and maximum injection time of 60 ms.

## Data processing and basic analysis

Raw data was imported into the Expressionist software (Genedata, San Francisco, CA) and processed as described previously (*Shalit et al., 2015*). The software was used for retention time alignment and peak detection of precursor peptides. A master peak list was generated from all MS/MS events and sent for database searching using Mascot v2.5 (Matrix Sciences) and MSGF+ (Integrative Omics, https://omics.pnl.gov/software/ms-gf). Data was searched against the Saccharomyces cerevisiae (strain ATCC 204508/S288c) protein database as downloaded from UniprotKB (http://www.uniprot.org/), and appended with 125 common laboratory contaminant proteins as well as the mCherry protein sequence (Uniprot accession X5DSL3). Fixed modification was set to carbamidomethylation of cysteines and variable modifications were set to oxidation of methionines and deamidation of N or Q. Search results were then filtered using the PeptideProphet algorithm (*Keller et al., 2002*) to achieve maximum false discovery rate of 1% at the protein level. Peptide identifications were imported back to Expressionist to annotate identified peaks. Quantification of proteins from the peptide data was performed using an in-house script (*Keller et al., 2002*). Data was normalized based on the total ion current. Protein abundance was obtained by summing the three most intense, unique peptides per protein. A Student's t-Test, after logarithmic transformation, was used to identify significant differences across the biological replica. If not annotated differently in the figure caption, the analysis was based on the median of three biological repeats.

## Proteomic datasets

We analyzed the following external mass-spectrometry datasets: two carbon sources (*Paulo et al., 2015*, *2016*) and diauxic shift dataset (*Murphy et al., 2015*). In each dataset, we normalized our glucose condition (SC) or time zero to the literature (median of 13 rich conditions from PaxBD (*Wang et al., 2012*), see conditions in *Figure 1—figure supplement 1B*, 'Rich Conditions'). For the analysis, we used only the genes also present in our dataset. The carbon source data set (*Paulo et al., 2015*, *2016*) growth rates were acquired from the papers or via personal communication. The diauxic shift data set (*Murphy et al., 2015*) growth rates were calculated from *Figure 4A* upper panel by applying a growth curve to the data.

## Gene groups

We divided the proteome into 12 groups, 11 of which were based on SGD GO annotations or KEGG annotations, which together account for 80% of the proteome (by protein abundance). The rest were grouped together as an additional 12[th] group. There is a small overlap between the groups, See *Supplementary file 1* for the protein names in each group.

## RNAseq transcription protocol and analysis

As described in *Voichek et al. (2016)*. The analysis was based on the median of 6–8 biological repeats (SC/Low N – 8; Low Pi – 6).

## Published RNA datasets

The different carbon datasets (*Figure 2B*,upper panel - dark orange, *Figure 2—figure supplement 1*) were obtained from (*Gasch et al., 2000*). Data for strains growing on xylulose (*Figure 2B*, upper panel - yellow) were obtained from (*Tamari et al., 2014*; *2016*). We compared ribosomal fractions from the RNAseq data with ChIP data between the two papers, and excluded from the analysis strains with high variability. Deletion strains (*Figure 2B*, bottom panel) datasets were obtained from (*Kemmeren et al., 2014*; *O'Duibhir et al., 2014*).

## RNA data of cells growing in low Pi media (0.06 mM)

Logarithmic WT cells ($\approx$0.4 OD) grown in SC medium were washed in low Pi (0.06 mM) media. Cells were then diluted and inoculated into fresh low Pi media to an initial OD of 0.05. Cells were grown

at 30°C for several hours. At every time point, as indicated at *Figure 4—figure supplement 1A*, a sample was extracted and RNAseq was performed as described above.

## Ribosomal fraction analysis in the RNA data

The ribosomal fraction was calculated by dividing the number of mRNA reads of ribosomal proteins by the total reads. In the ChIP data, the ribosomal fraction was calculated as the relative amount normalized to the SC condition (Logarithmic mean).

## mCherry level measurements

The mCherry levels were calculated from our mass-spectrometry data. To compare the mCherry between the different copy numbers, we used the relative mass-spectrometry data. In order to measure our reference strain (one copy in SC), we used TDH3p-mCherry (protein fusion) strain as described in the Strains section. The amount of mCherry in this strain is equal to the TDH3 protein ($\approx$ 2.2% base on the iBAQ parameter). The mCherry level of our one mCherry copy strain was 14.5% lower than the mCherry fluorescence of the TDH3p-mCherry strain. Thus, the mCherry level of one copy strain is $\approx$ 1.9% of the proteome. Notably, this calibration is in very good agreement with our previous fluorescence-based calibration (*Kafri et al., 2016b*).

## Chemostat experiments

The chemostat experiments were performed using a DASBox Bacterial Fermentation system (DASGIP, Eppendorf). Yeast cells containing one copy number were grown in the fermenter at 30°C in SC with low Pi. The fermentation started with the inoculation of an exponentially growing starter at $OD_{600}$0.5–1 diluted x10. Culture was initially grown in a batch mode for 6–8 hr to an $OD_{600}$ between 0.5–1. The culture was then switched to chemostat mode with final doubling time of 3 hr/doubling. When the culture reached steady state in the chemostat, we spiked in the same media containing high levels of phosphate for three hours.

## Live microscopy environment perturbation experiment

Cells were grown in a standard FCS2 flow cell (Bioptechs) improved similar to (*Charvin et al., 2010*). Cells growing in the flow cell were imaged with Olympus IX81 microscope with automated stage and ZDC autofocus and cooled CCD camera (Hamamatsu). In order to grow the cells in a planar layer while simultaneously control their extra cellular environment, standard FCS2 flow cell (Bioptechs) was used and improved in the following way (*Charvin et al., 2010*): the 40 mm round cover slips were coated with a thin layer of PDMS (Sylgrad 184, GE). This was done in a clean room, using a suitable spinning procedure to achieve layers of $\approx$ 30 μm thick. To confine the cells and prevent their movement, while medium is flowing through the chamber (*Balaban et al., 2004*), a diffusive cellulose membrane, which was cut from dialysis tubes (Sigma Aldrich, D9527) was used.

The membrane was cleaned and prepared as described in *Charvin et al., 2010*. The cells are confined between the PDMS layer and the membrane, while the medium is flowing above the membrane. In this way the medium reaches the cells area without exerting too much force on the cells. The flow cell is then connected to two micro-perfusion pumps (Instech), which are controlled automatically by the computer via a D/A convener (measurement computing usb-3110), enabling the exchange of medium during the experiment without perturbing the cells. The media tanks for the environment perturbation experiment were continuously stirred, to allow for better aeration of the medium. The whole system is controlled by ImagePro 6.3.1 (Media Cybernetics).

Galactose to glucose shift experiments were done by replacing synthetic complete medium based on 2% galactose (SC-Gal) with synthetic complete medium based on 2% glucose (SC-Glu). Cell volume was estimated from the bright field images assuming that the yeast cells are prolate spheroids.

## Polysome profiling

100 ml cultures of exponentially growing yeast cells ($OD_{600}$0.2–0.4 with at least 6 exponential divisions since their dilution) were treated with 100 μg/ml Cycloheximide for 5 min at 30°C, harvested, and then washed with cold Buffer AB (20 mM Tris-HCl pH 7.4, 50 mM KCl, 10 mM MgCl2, 1 mM DTT and 100 μg/ml Cycloheximide). The cells were lysed with 250 μl Buffer AB using glass beads as

in (*Pospísek and Valásek, 2013*). Cleared lysates were loaded onto 10–50% sucrose gradients with the same AB buffer composition containing 100 µg/ml Cycloheximide, and centrifuged at 39,000 RPM in a SW41 rotor for 2.5 hr at 4°C. Gradients were continuously recorded using Biocomp gradient station at $OD_{254}$ nm.

Profiles were aligned in the x-axis by their peak locations and in the y-axis to SC low burden. A blank sucrose gradient measurement was subtracted from all the samples. Next, the integrals of the 40S, 60S, monosomes and polysome fractions (disomes and higher) were measured. The ratio of polysomes to monosomes was defined as the (polysomes integral) / (40S + 60S + monosomes integrals). Both mRNA and rRNA are present in a given mono/polysome signal measured by the $OD_{254}$ nm absorbance. The total length of rRNA in one ribosome is about 5500 bp while the average mRNA length is 1250 bp (*Milo et al., 2010*). Accordingly, the rRNA fraction of total RNA in monosomes is ~82%, and in polysomes containing 5 ribosomes is ~96%. This differential proportion was taken into account when calculating the inactive ribosome fraction.

## Interpreting polysomal profiles

Our analysis of the polysomal profiles (*Figure 3* and *Figure 6*) assumes that the monsome fraction represents the inactive ribosomes.

The common notion is that the majority of monosomes represent inactive ribosomes that are not actively translating (*Aspden et al., 2014*; *Castelli et al., 2011*; *Chassé et al., 2017*; *Kapp et al., 2004*; *Kelen et al., 2009*; *Liu and Qian, 2016*; *Warner et al., 1963*). This notion was recently challenged by a study which analyzed monosomes using ribosome profiling, identifying some translation in the monosome peak, although the fraction of monosomes that is actively translating remained unclear (*Heyer and Moore, 2016*). Of note, monosomal ribosomes were shown to be present primarily in the vicinity of the translation start site, with only a minor fraction spread across the full gene, in sharp contrast to the ribosomes in polysomes which were generally distributed evenly across the full ORF, as expected from a primarily elongating species (Heyer and Moore, 2016, *Figure 2*). This supports the common notion that, at any given instance, the majority of monosomes do not actively contribute to protein production.

## Literature values for translation elongation rates

Direct comparisons of translation elongation rates of cells growing in different conditions were reported in two papers: *Waldron et al. (1977)* and *Bonven and Gulløv (1979)*. These papers reached different conclusions: *Waldron et al. (1977)* directly measured translation elongation rates of an active translating ribosome and concluded that translation elongation rates do not change between rich (SC) and low-nitrogen media, with a constant value of 10.5 aa/sec measured in both conditions, despite significant differences in cell growth rate. *Bonven and Gulløv (1979)* used basically the same method but reached a different conclusion, reporting 40% decrease in elongation rate between the growth on glucose and on acetate (with values of 9.6 aa/sec to 5.4 aa/sec, respectively). The difference between the two reports could be due to the use of different media or alternatively, details of the experiments. For example, *Waldron et al. (1977)* quantified translation rate using proteins of high molecular weights (mentioning larger errors in the short-time estimates), while (*Bonven and Gulløv, 1979*) estimated elongation rates of the slow-growing cells focusing on three subclasses of small molecular weight proteins, (while noting that an "*increase in the uptake rate is registered at 70 s*", perhaps indicating that elongation rate at the higher molecular weight regime are closer to the glucose-measured values). *Boehlke and Friesen, 1975* also reported translation elongation rates, concluding that the translation elongation rate change with growth rate. These authors, however, did not measure elongation rates directly, but in fact calculated this value by measuring ribosome content and growth rate, assuming that ~90% of the expressed ribosomes are active in each media. This study, however, did not detect growth-rate dependent changes in the ribosome expression level or in the polysome profiles, in disagreement with broad available literature. The authors' estimates are therefore based on the assumptions that both the ribosome content and the active fraction are independent of growth rate, which is unlikely to hold.

## Data availability

The RNAseq data is available via accession number: SRP107059

## Acknowledgements

We thank J Paulo for sending unpublished growth data, G Jona for assistance with the chemostats and our lab members for fruitful discussions. We thank Roni Winkler and Noam Stern-Ginossar for help and instruments used for polysome profiling. We would like to thank Dr. Yishai Levin, Dr. Alon Savidor and Dalia Elinger of the INCPM Institute for protein profiling for handling, running and initial analysis of the mass spectrometry samples. This work was supported by the ERC, the ISF and by the Minerva Center.

## Additional information

### Competing interests

Naama Barkai: Senior editor, eLife. The other authors declare that no competing interests exist.

### Funding

| Funder | Author |
| --- | --- |
| Israel Science Foundation | Naama Barkai |
| H2020 European Research Council | Naama Barkai |

The funders had no role in study design, data collection and interpretation, or the decision to submit the work for publication.

### Author contributions

Eyal Metzl-Raz, Conceptualization, Resources, Data curation, Software, Formal analysis, Validation, Investigation, Visulaization, Methodology, Writing—original draft, Writing—review and editing; Moshe Kafri, Conceptualization, Resources, Data curation, Software, Formal analysis, Validation, Investigation, Visualization, Methodology, Writing—original draft, Writing—review and editing; Gilad Yaakov, Resources, Data curation, Formal analysis, Methodology, Writing—review and editing; Ilya Soifer, Resources, Software, Writing—review and editing; Yonat Gurvich, Resources, Data curation; Naama Barkai, Conceptualization, Supervision, Funding acquisition, Investigation, Methodology, Writing—original draft, Project administration, Writing—review and editing

### Author ORCIDs

Eyal Metzl-Raz https://orcid.org/0000-0002-7111-2822

Naama Barkai https://orcid.org/0000-0002-2444-6061

### Decision letter and Author response

Decision letter https://doi.org/10.7554/eLife.28034.020

Author response https://doi.org/10.7554/eLife.28034.021

## Additional files

### Supplementary files

• Supplementary file 1. Proteome functional groups: We divided the proteome into 12 groups, 11 of which were based on SGD GO annotations or KEGG annotations, which together account for 80% of the proteome (by protein abundance). The rest were grouped together as an additional 12th group.
DOI: https://doi.org/10.7554/eLife.28034.014

• Supplementary file 2. Proteins group used for *Figure 1B*.
DOI: https://doi.org/10.7554/eLife.28034.015

• Supplementary file 3. One copy mCherry calculation: LC-MS/MS proteomic data for one genomic copy of mCherry and for fused mCherry (see Materials and methods).
DOI: https://doi.org/10.7554/eLife.28034.016

• Transparent reporting form

DOI: https://doi.org/10.7554/eLife.28034.017

### Major datasets

The following dataset was generated:

| Author(s) | Year | Dataset title | Dataset URL | Database, license, and accessibility information |
| --- | --- | --- | --- | --- |
| Eyal Metzl-Raz, Moshe Kafri, Gilad Yaakov, Ilya Soifer, Yonat Gurvich, Naama Barkai | 2017 | Saccharomyces cerevisiae strain: Y8205 Raw sequence reads | https://www.ncbi.nlm.nih.gov/sra/?term=SRP107059 | Publicly available at the Sequence Read Archive (accession no. SRP107059) |

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
