## [Decision Letter]

Thank you for submitting your article "Principles of cellular resource allocation revealed by condition-dependent proteome profiling" for consideration by *eLife*. Your article has been reviewed by three peer reviewers, and the evaluation has been overseen by a Reviewing Editor and Patricia Wittkopp as the Senior Editor. The reviewers have opted to remain anonymous.

The reviewers have discussed the reviews with one another and the Reviewing Editor has drafted this decision to help you prepare a revised submission.

Summary:

This manuscript describes a crucial relationship between the ribosome population and growth rate, controlled by nutrient conditions. It seeks, in a model eukaryote, to extend findings made in *E. coli* and long-standing in the literature, that the rRNA content of a cell is linearly dependent upon growth rate. However, the manuscript seeks to further extend these findings in yeast, by identifying the existence of 'reserve' ribosomal capacity, which the authors argue is used in cases of rapid nutrient upshift. The manuscript discoveries in yeast confirm similar findings made in *E. coli*, that reserve ribosome capacity exists in *E. coli* grown at slow growth rates. We are all enthusiastic about the potential of this work to be suitable for *eLife*, but also have some comments/questions that need to be addressed before moving forward.

Essential revisions:

1) The discovery that 8% of the proteome encodes excess ribosomal proteins, not actively employed in translation, depends on a quantitative scaling, and an interpretation of the author's central growth relationship, namely that;

Growth rate = Γ (translation elongation rate) * [ribosome active]

And by extension, since ribosome(sum) = ribosome(active) + ribosome(inactive);

Ribosome(sum) = mu/γ + ribosome(inactive)

The authors make the assumption that the r0 inactive ribosome fraction is invariant, then the rate of translation is also invariant. However, r0 and the rate of translation are both variables in the central equation governing these experiments, thus unless it is certain that the elongation rate is invariant with changes in growth rate, the assumptions made in fact break down.

Of direct relevance to this point, there is good evidence in the literature that crucially, γ, the translation elongation rate is proportional to growth rate, a relationship governed apparently by a linear function. This discovery is reported in at least two publications, PMID: 372763 and PMID: 1089627, not cited in the paper, but whose findings impact directly on one of the central assumptions of the paper. Indeed, using the discovery that the translation elongation rate scales with growth rate, Bonven and Gullen reach the following conclusion, at odds with the conclusions reached in this manuscript;

Bonven and Gullen, Mol Gen Genet. 1979;170:225-30 "When our Cp estimations are taken into account (Table 1) this pattern can be interpreted to mean that the number of active ribosomes decreases drastically with decreasing growth rate, resulting in an overproduction of ribosomes at low growth rates. The vast amount of idled ribosomes observed might be seen throughout the cell cycle or could be restricted to discrete steps within the cell cycle." [note the Cp that Bonven and Gullov refer to is the peptide chain elongation rate)

Bonven and Gullen adopt the same analytic approach to defining the relationship between ribosome content and growth rate, and also examine the effect of nutrient upshifts on this relationship.

Further, in this manuscript under review, the authors also acknowledge that their monosome estimates, used in part to judge the proportion of inactive ribosomes, may be subject to error, since as they state, "In contrast, monosomes are thought to contain primarily inactive ribosomes, although some of these may represent low-translating genes (Aspden et al., 2014; Heyer and Moore, 2016; Kelen et al., 2009)". This is a material observation when the fraction on idling ribosomes they estimate is relatively small, and subject to experimental measurement variation.

In summary, there are important factors not taken into account in the authors' analysis which may materially change the conclusions they reach. Since these factors include assumptions made directly at odds with other literature reports, addressing these in full is critical to make the manuscript suitable for publication.

2) The reported specific growth rate goes to above 0.6 h-1. I have never seen so high values for *S. cerevisiae* before, and I doubt these values can be true. Not because this is important for the story as it will probably be a linear mistake in the estimation of this value, and hence all the values reported are too high. The authors refer to their previous paper for calculation of this, but that does not help me much when checking. So this should be addressed and the manuscript should be revised. The problem will be that if these numbers are not corrected there will be a lot of confusion in referring to data from this paper and comparing the data here with data from other studies.

3) The fraction of mRNA transcripts that code for ribosomal proteins as a function of cell growth rate showed the similar quantitative scaling as at the ribosomal fraction at the of the proteome level. Considering a simple steady-state scenario where protein level corresponds to the mRNA level multiplied by the ratio of translation to mRNA degradation, this observation suggest that this ratio is similar for all protein. It would be insightful to address this point and explain whether it is reasonable.

4) In using area under the curve of polysome profiling, it is important to mention that the area is proportional for both the ribosome and the mRNA that it is bound to. This might lead to overestimation of the "nonactive" ribosome. As this is an important experimental validation to the inferred active ribosome fraction, it should be taken into account at least as a margin of error.

5) The authors suggest that cells entering stationary phase transiently decrease their predicted fraction of inactive ribosomes (r0). The authors should explain why the non-linear relations in Figure 4 is the result of changing r0 and not the translation rate.

6) When forcing cells to produce more proteins, the authors examine the change of growth rate for same burden but different media conditions. This relation gives a slope which is similar. One can also consider the change in growth for the same condition but different burdens. In a previous figure the author showed that different strains that grow on xylose (one way to manipulate growth) and different conditions (second way to manipulate growth) falls on the same curve. In that case the change of growth rate has a different slope for different conditions which suggest that the translation rate changes. In other words, the relation between growth rate and r is not unique and does depends on the way growth rate is changed. The authors should address this point. This is also relevant to the results presented in Figure 6. The parameters could be inferred from the slopes within burden and not within condition.

7) The logic behind the relation between growth rate and r (the amount of protein that is produced during doubling is related to the translation rate multiply by the active ribosomes) seems to hold also the double mutant data in Figure 2. It seems that slope is smaller than the condition dependent case but it is not zero. What does this smaller slope mean? What does the bigger interaction mean? This is important in the context of using this observation to support the suggestion that growth rate feedback is less dominant as suggested in Figure 7.

---

## [Author Response]

*Essential revisions:*

*1) The discovery that 8% of the proteome encodes excess ribosomal proteins, not actively employed in translation, depends on a quantitative scaling, and an interpretation of the author's central growth relationship, namely that;*

*Growth rate = Γ (translation elongation rate) * [ribosome active]*

*And by extension, since ribosome(sum) = ribosome(active) + ribosome(inactive);*

*Ribosome(sum) = mu/γ + ribosome(inactive)*

*The authors make the assumption that the r0 inactive ribosome fraction is invariant, then the rate of translation is also invariant. However, r0 and the rate of translation are both variables in the central equation governing these experiments, thus unless it is certain that the elongation rate is invariant with changes in growth rate, the assumptions made in fact break down.*

Our conclusion that r_0_ is invariant to growth rate is based on the following:

1. Experimentally, we observe a linear scaling between growth rate (μ) and the ribosomal protein fraction (*r*) (Figure 2). Given the theoretically-expected relation, cited by the reviewer above, this scaling implies a tight coordination between the two potentially-varying parameters g and r_0_.

2. In the simplest scenario that maintains the experimentally-observed linear scaling, both γ and *r_0_* remain constant, independent of growth rate. This would lead to the observed linear scaling without a need for any further fine-tuning.

3. To experimentally test whether r_0_ changes with growth rate, or whether it remains invariant, we used polysome profiling. Our assumption here is that the ratio of polysomes to monosomes well estimates the ratio of active to inactive ribosome, *(r-r_0_)/r_0_* (see below). Together with the measure total ribosomal fraction, *r* the monosome-to-polysome ratio therefore provides a direct test for *r_0_*. Indeed, this measure is quantitatively consistent with a constant *r_0_*~8%, the same value we obtained at the limit of no-growth rate (Figure 3). This experiment therefore provides a direct test to our conclusion of invariant *r_0._*

We agree with the reviewers that this rational wasn’t well explained and have therefore rewritten the relevant sections:

“Quantitative interpretation of the scaling between ribosome content and growth rate: The scaling relation between ribosome content, r, and the specific growth rate, *μ*,is typically interpreted based on the general relationship between protein translation and growth rate during balanced growth. […]The measured scaling curve, Figure 2, however, implies that these parameters are tightly coordinated to maintain the linear relationship between ribosome content and growth rate across the different conditions.”

Following this paragraph, which introduces the linear scaling, the model and its implications for the connection between g and r_0_, we discuss the different possibilities of interpreting this scaling, cite literature, and present the polysome profiling as means for directly testing whether *r_0_*remains constant. This is detailed below, as it relates more directly to the next set of comments.

*Of direct relevance to this point, there is good evidence in the literature that crucially, γ, the translation elongation rate is proportional to growth rate, a relationship governed apparently by a linear function. This discovery is reported in at least two publications, PMID: 372763 and PMID: 1089627, not cited in the paper, but whose findings impact directly on one of the central assumptions of the paper.*

The missing references were added to the manuscript.

Reference PMID: 1089627 (“cellular content of ribonucleic acid and protein in *S. cerevisiae* … calculation of an apparent peptide chain elongation rate”) does not measure the elongation rates of a single active ribosome (aa/sec) but in fact calculates this value by measuring ribosome content and growth rate, and assuming that ~90% of the expressed ribosomes are active in each media. It is interesting to note that this study did not detect growth-rate dependent changes in the ribosome expression level, or in the polysome profiles, in disagreement with broad available literature. The authors therefore assume growth-rate independent ribosome content and similarly growth-rate independent active fraction, and calculated an apparent translation elongation rates using these assumptions, which we now know does not hold.

References PMID: 343781 (which we cited) and PMID: 372763 (which we now cite) are perhaps more relevant as they measured elongation rates directly. These two papers have reached different conclusions.

PMID: 343781 directly measured translation elongation rates of an active translating ribosome. These experiments concluded that translation elongation rates do not change between rich (SC) and low-nitrogen media, with a constant value of 10.5 aa/sec measured in both conditions, despite a significant difference in cell growth rate.

PMID: 372763 used the same method as PMID: 343781 but reached a different conclusion. We have missed this paper before and thank the reviewer for bringing it to our attention. In this paper, translation elongation rates were measured in rich and low acetate media, reporting 40% decrease in elongation rate between the two media (from 9.6 aa/sec to 5.4 aa/sec). We are not sure why these two papers reached different conclusions. It could be the result of using different media or, alternatively, details of the experiments might have been different. For example, we noted that while the paper above reporting invariant translation rate (PMID:372763) quantified translation rates using proteins of high molecular weights (mentioning larger errors in the short-time estimates), the second paper reporting condition-dependent rates (PMID:343781) estimated elongation rates in slow-growing cells focusing on this precise short-time limit, considering three subclasses of small molecular weight proteins, (while noting that an *“increase in the uptake rate is registered at 70 s”*, perhaps indicating that elongation rate at the higher molecular weight regime are closer to the glucose-measured values).

Given this discrepancy between the two papers, we now rephrased the way we cite previous measurements of elongation rates, specifically noting this disagreement in the literature. This further motivates our use of polysome profiling for estimating the residual fraction of inactive ribosomes.

More specifically, we now relate explicitly to the two possibilities: a constant elongation rate and elongation rate that scales with growth rate, as suggested by the reviewer. Note that these two models provide different predictions for the residual fraction of inactive ribosomes: *r_0_≈8%* in the case of constant elongation rate (as we wrote) and *r_0_ = r-a, with a* being some constant for elongation rates that scales with growth rate. Accordingly, the ratio of inactive to active ribosome (*r_0_/(r- r_0_*) increases with decreasing growth rate in the first scenario, but remains constant in the second scenario. Our polysome profiling is clearly in quantitative agreement with the first possibility of invariant *r_0_*

The relevant paragraphs are now written as follows:

“In our analysis above, we distinguished the fraction of ribosomal proteins that contribute to cell growth (*μ=γ r_a_*) from the residual inactive, non-translating part (*r_0_*). […] Therefore, the fraction of ribosomes that are not actively translating ranges from ≈25% in rapidly growing cells (*r_0_*≈8% relative to the total ribosomal fraction r≈30%) and further increases with decreasing growth rate (Figure 3).”

*Indeed, using the discovery that the translation elongation rate scales with growth rate, Bonven and Gullen reach the following conclusion, at odds with the conclusions reached in this manuscript;*

*Bonven and Gullen, Mol Gen Genet. 1979;170:225-30 "When our Cp estimations are taken into account (Table 1) this pattern can be interpreted to mean that the number of active ribosomes decreases drastically with decreasing growth rate, resulting in an overproduction of ribosomes at low growth rates. The vast amount of idled ribosomes observed might be seen throughout the cell cycle or could be restricted to discrete steps within the cell cycle." [note the Cp that Bonven and Gullov refer to is the peptide chain elongation rate)*

*Bonven and Gullen adopt the same analytic approach to defining the relationship between ribosome content and growth rate, and also examine the effect of nutrient upshifts on this relationship.*

Please see our reply above: Bonven and Gullen, Mol Gen Genet. 1979 is the PMID 372763 which we refer to above, reporting elongation rates vary with growth conditions. This is in contrast to PMID: 343781 which reported condition-invariant elongation rate. Our results are consistent with later study, but not with the former, as we acknowledge above and in the manuscript.

(Please also note that, despite this disagreement regarding elongation rate, also PMID 372763 concluded that the fraction of inactive ribosomes largely increases with decreasing growth rate, namely that cells function far from the growth-optimizing limit where all ribosomes are fully employed in translation.)

*Further, in this manuscript under review, the authors also acknowledge that their monosome estimates, used in part to judge the proportion of inactive ribosomes, may be subject to error, since as they state, "In contrast, monosomes are thought to contain primarily inactive ribosomes, although some of these may represent low-translating genes (Aspden et al., 2014; Heyer and Moore, 2016; Kelen et al., 2009)". This is a material observation when the fraction on idling ribosomes they estimate is relatively small, and subject to experimental measurement variation.*

The common notion is that most monosomes represent inactive ribosomes not actively translating. In fact, in our survey of the literature, we found and accordingly cite, only one paper which challenges the interpretation of monosomes as non-translating ribosomes (Heyer and Moore, 2016). Notably, while this last paper provided evidence for some translation in the monosome peak (and characterized their properties), it lacks information about the fraction of monosomes that is actively translating. Furthermore, the data in this paper clearly shows a large difference in translation efficiency of the ribosomes present in monosomes vs. polysomes: thus, while ribosomes in polysomes are generally distributed evenly across the full ORF, as expected from a primary elongating species, monsomal ribosomes are primarily in the vicinity of the translation start site, with only a minor fraction spread across the full gene (Heyer and Moore, 2016, Figure 2). This supports the common notion that, at any given instance, most monosomes do not actively contribute to protein production.

The following paragraph was added to subsection “Interpreting polysomal profiles”:

“Our analysis of the polysomal profiles (Figure 3 and Figure 6) assumes that the monsome fraction represents the inactive ribosomes.

The common notion is that the majority of monosomes represent inactive ribosomes that are not actively translating (Aspden et al., 2014; Castelli et al., 2011; Chassé et al., 2017; Kapp et al., 2004; Kelen et al., 2009; Liu and Qian, 2016; Warner, Knopf and Rich, 1963). […] This supports the common notion that, at any given instance, the majority of monosomes do not actively contribute to protein production.”

*In summary, there are important factors not taken into account in the authors' analysis which may materially change the conclusions they reach. Since these factors include assumptions made directly at odds with other literature reports, addressing these in full is critical to make the manuscript suitable for publication.*

We thank the reviewer for the insightful comments and hope that our clarification and corrections to the manuscript addressed the concerns.

*2) The reported specific growth rate goes to above 0.6 h-1. I have never seen so high values for S. cerevisiae before, and I doubt these values can be true. Not because this is important for the story as it will probably be a linear mistake in the estimation of this value, and hence all the values reported are too high. The authors refer to their previous paper for calculation of this, but that does not help me much when checking. So this should be addressed and the manuscript should be revised. The problem will be that if these numbers are not corrected there will be a lot of confusion in referring to data from this paper and comparing the data here with data from other studies.*

We thank the reviewer for this comment. The highest growth rate we measured corresponds to a division time of ≈1.5 hours, which is similar to commonly observed growth rates in YPD [e.g. (Sherman, 2002)].Generally, reported growth rate in SC are somewhat lower, ranging between 0.38-0.54 h^-1^. We are confident about the growth rates we report as we are routinely measuring this in our lab, using the same SC media prepared for us in the Weizmann’s biological service unit.

This comment, however, made us realize that we did not in fact plot growth rate μ [μ=ln(2)/T^div^] but the inverse of division times (1/T^div^). We now annotated the axis correctly as Generations per hour.

Accordingly, our comparison with the estimated ribosomal elongation rate g had to be revised. The addition of ln(2) factor to the measured slope made it 30% smaller than the theory-predicted ribosome translation elongation rate. This discrepancy may reflect uncertainty in estimating γ: for example, this parameter depends on the number of proteins (and total amino-acid length) that needs to be translated to make a functional ribosome, which may include, in addition to the ‘core’ of ribosomal proteins, also additional translational factors. Alternatively, the simplified theory may still be lacking in details, such as delays in the kinetics of ribosome production. It is interesting that the measured slope would in fact have matched the predicted slope if ribosome production during each individual cell cycle was linear, rather than exponential, since in this case, the effective growth rate would be given by 1/T^d^ rather than ln(2)/ T^d^. We are aware, of course, that growth is likely to be exponential, but still find it interesting enough to be highlighted in the text.

Interestingly, following our realization of this log(2) difference between the measured and predicted slope, we re-examined the study of Scott et al. (Scott et al., 2010), which analyzed the analogous scaling in bacterial. Interestingly, these authors also needed to scale the ribosome fraction by a similar factor [1.67≈ 1/ln(2)] in order to achieve the reported agreement between the measured and estimated slope. This is noted in their Supplementary Information: “…where M_R_ is the total mass of the ribosomal proteins together with their affiliates (referred to below as the “extended ribosome”), including all the initiation factors, elongation factors, tRNA synthases, etc.”. The fact that log(2) factor is required to achieve agreement between the measured and calculated slope in both yeast and bacteria is intriguing, although we are not sure about its interpretation.

We now refer to this discrepancy in the main text

“We can estimate *γ* using measured parameters: the translation elongation rate (10.5 aa/sec) (Waldron, Jund and Lacroute, 1977)and the number of amino-acids composing the budding yeast ribosome (12,485 A.A). […] In this case, the relation between the specific growth rate μ and the division time T_d_ is *μ=*1/T_d_ rather than *μ=*ln(2)/T_d_. “

And also in the Discussion:

“We find that also in budding yeast, the proteome fraction coding for ribosomal proteins scales linearly with the specific growth rate. […]Alternatively, studies in bacteria attributed this difference to the need for including additional proteins for production of functional ribosomal unit, which may change the predicted value of the slope (Klumpp et al., 2013; Scott et al., 2010).”

We would like to emphasize that the scaling curve plotted in Figure 2 was derived directly from the data and required no tuning of parameters. For example, the group of ribosomal proteins was defined to include all genes annotate directly to the ribosome, as listed in Supplementary file 1. Further, as we showed in Figure 1 top, we also extended this group to include all additional factors annotated as translation factors, which, in rich media, occupied ~40% of the proteome. Abundance of this group also decreased linearly with growth rate, with practically the same slope as the core ribosomal group (Figure 2).

*3) The fraction of mRNA transcripts that code for ribosomal proteins as a function of cell growth rate showed the similar quantitative scaling as at the ribosomal fraction at the of the proteome level. Considering a simple steady-state scenario where protein level corresponds to the mRNA level multiplied by the ratio of translation to mRNA degradation, this observation suggest that this ratio is similar for all protein. It would be insightful to address this point and explain whether it is reasonable.*

We agree. More precisely, to obtain the measured agreement, the ratio of translation/degradation of ribosomal proteins should be the same as the mean ratio of translation/degradation of all other proteins. This is indeed conceivable: Most highly abundant proteins are stable, and therefore degradation rates are controlled primary by growth-dependent dilution. Our results therefore imply that the average translation rate of ribosomal proteins is the same as the average translation rates of all other proteins (primarily highly abundant ones). We now refer to this in the caption to Figure 2.

Note the high agreement between the transcriptomic data and the proteomic linear fit, implying that on average, the translation-to-degradation of ribosomal proteins is the same as that of other proteins (primarily highly abundant ones)

*4) In using area under the curve of polysome profiling, it is important to mention that the area is proportional for both the ribosome and the mRNA that it is bound to. This might lead to overestimation of the "nonactive" ribosome. As this is an important experimental validation to the inferred active ribosome fraction, it should be taken into account at least as a margin of error.*

We agree. We now added the mRNA mass to our calculation.

Specifically, the area under the curve reflects the RNA content. The total length of rRNA in one ribosome is about 5500 bp, and the average mRNA length is 1250bp (Milo et al., 2010). Accordingly, the rRNA fraction of total RNA in monosomes is ~81.5%, and in polysome containing e.g. 5 ribosomes is ~95.6%. We now use those normalizations in the figures and add it to the Materials and methods. These corrections have only minor effects on the calculated inactive ribosome fraction (ρ): in SC it reduced from 0.2 to 0.18 and in low N from 0.4 to 0.37.

*5) The authors suggest that cells entering stationary phase transiently decrease their predicted fraction of inactive ribosomes (r0). The authors should explain why the non-linear relations in Figure 4 is the result of changing r0 and not the translation rate.*

We agree that our data does not exclude the possibility that elongation rate, rather than r_0_ is transiently changing. We find it unlikely, given the constancy of translation elongation rate across conditions, but on the same token, we agree that r_0_ is also constant across conditions. Indeed, for this reason we have further explored the possibility of decreasing *r_0_*by forcing protein expression, in which case we believe our data is conclusive on that point.

We have rephrased the two relevant sections. Section titles now read:

“The scaling between ribosome content and growth rate changes when cells prepare to enter stationary phase”

“The scaling between ribosome content and growth rate changes during nutrient upshift”

The conclusion parts in the sections themselves were also rephrased:

“…This could be interpreted in two ways: a transient increase in translation elongation rate, or a more efficient use of available ribosomes, leading to lower *r_0_*.”

“… Again, this could be interpreted in two ways: either cells transiently increase the translation elongation rate, or they make a more efficient use of the already available ribosomes by decreasing the non-translating fraction *r_0_*.”

“… Therefore, cells increase their growth rate, and thus protein production rates, within a short time that appears insufficient for synthesizing new ribosomes, consistent with either a transient increase in translation elongation rate, or a more efficient use of available ribosomes through a decrease in the non-translating fraction *r_0_*.”

*6) When forcing cells to produce more proteins, the authors examine the change of growth rate for same burden but different media conditions. This relation gives a slope which is similar. One can also consider the change in growth for the same condition but different burdens. In a previous figure the author showed that different strains that grow on xylose (one way to manipulate growth) and different conditions (second way to manipulate growth) falls on the same curve. In that case the change of growth rate has a different slope for different conditions which suggest that the translation rate changes. In other words, the relation between growth rate and r is not unique and does depends on the way growth rate is changed. The authors should address this point. This is also relevant to the results presented in Figure. 6. The parameters could be inferred from the slopes within burden and not within condition.*

We agree: this finding – that the relationship between growth rate and *r* is not unique but depends on the way by which growth rate is changing – is one of our central results. As mentioned by the reviewer, this result implies that the scaling curve we observe is not an inherent unavoidable consequence of cell physiology, but is a consequence of signaling which sets the fraction of expressed ribosomes, as well as the ratio of active vs. inactive ribosomes as a function of growth conditions.

We highlight this point in the Discussion:

“We provided evidences that a constant fraction (≈8%) of the budding yeast proteome encodes for ribosomal proteins that are not actively translating at a given time. […]These results implies that the scaling curve we observed, and in particular the fraction of residual, inactive ribosomes, is not an inherent unavoidable consequence of cell physiology or the mechanism of ribosome production, but is rather a consequence of signaling which sets the ratio of active vs. inactive ribosomes as a function of growth conditions.”

More specifically, the scaling between ribosome content and growth rate depends on two parameters: the translation elongation rate (γ) or the fraction of inactive ribosome *r_0._*When comparing the same strain growing in different conditions, our analysis concluded that both of these parameters, g and *r_0_,* remain constant, as we discuss in our answers to the first two comments, above. The fact that the same slope and same residual is observed also when comparing different strains growing in one media (xylulose), suggests that also here, all strains have the same elongation rate g and express the same residual ribosomal fraction *r_0_*but differ only in the fraction of expressed ribosome *r*.

The question is what happens when we add burden to the cells, namely force them to express high amount of mCherry proteins. To answer that, we first looked separately at each individual burden strain, and examined the condition-dependent scaling curve for each such strain. What we find is a clear scaling, with burden-invariant slope, but a different intercept. Following our reasoning above, this is interpreted as a burden-invariant elongation rate g, coupled with a burden-dependent *r_0_.*

Therefore, our interpretation of the linear relation between ribosome content and growth rate obtained by changing burden levels, is that the residual inactive ribosomes, *r_0_* are reduced with the burden (inter-condition slope), while translation elongation rates remain unchanged (between condition, same burden slopes). In this situation the simplest mechanistic explanation is that *r_0_* is reduced proportionally to the forced mCherry expression.

We hope that this is now better explained in the manuscript.

*7) The logic behind the relation between growth rate and r (the amount of protein that is produced during doubling is related to the translation rate multiply by the active ribosomes) seems to hold also the double mutant data in Figure 2. It seems that slope is smaller than the condition dependent case but it is not zero. What does this smaller slope mean? What does the bigger interaction mean? This is important in the context of using this observation to support the suggestion that growth rate feedback is less dominant as suggested in Figure 7.*

The mutant data defines a rather noisy relation between ribosome content and growth rate. This is in sharp contrast to the almost noise-free relation observed when comparing different conditions. For this reason, we do not consider all mutant as showing one consistent behavior. Rather, it appears that each of the mutants breaks the relation between the ribosomal content and growth rate in a different way.

More specifically, the dominant signal in the mutant graph is that growth rate is reduced to a larger extent than do the ribosomal-protein transcripts. This would mean that cells continue to produce similar amount of ribosomal transcripts but grow more slowly due to other limitations imposed by the mutations. This could either imply that the mutants reduce the translation elongation rate, or more likely, that they over-produce ribosomes given their growth rate (they produce the ribosomes needed for the environmental expected growth rate and not for their actual growth rate), so that a larger fraction of the ribosomal proteins are now inactive.

The (relatively minor) decrease in ribosomal protein content in the mutant can be interpreted in two ways. First, it could indicate some internal feedback. The large variations between the mutants, however, would suggest that this feedback is rather weak. A second possibility is that some mutations also affect the production of ribosomal transcripts. This could be direct (e.g. mutations in regulators of these transcripts) or indirect (e.g. induction of stress response that reduces the fraction of ribosomal transcripts).

We now discuss these points in the manuscript.